# Integrative analysis of DNA replication origins and ORC-/MCM-binding sites in human cells reveals a lack of overlap

**Mengxue Tian[1,2], Zhenjia Wang[1], Zhangli Su[2,3], Etsuko Shibata[2,3], Yoshiyuki Shibata[2,3], Anindya Dutta[2,3]\*, Chongzhi Zang[1,2,4]\***

[1]Center for Public Health Genomics, University of Virginia School of Medicine, Charlottesville, United States; [2]Department of Biochemistry and Molecular Genetics, University of Virginia School of Medicine, Charlottesville, United States; [3]Department of Genetics, University of Alabama at Birmingham, Birmingham, United States; [4]Department of Public Health Sciences, University of Virginia, Charlottesville, United States

**\*For correspondence:**
Duttaa@uab.edu (AD);
zang@virginia.edu (CZ)

**Competing interest:** The authors declare that no competing interests exist.

**Abstract** Based on experimentally determined average inter-origin distances of ~100 kb, DNA replication initiates from ~50,000 origins on human chromosomes in each cell cycle. The origins are believed to be specified by binding of factors like the origin recognition complex (ORC) or CTCF or other features like G-quadruplexes. We have performed an integrative analysis of 113 genome-wide human origin profiles (from five different techniques) and five ORC-binding profiles to critically evaluate whether the most reproducible origins are specified by these features. Out of ~7.5 million union origins identified by all datasets, only 0.27% (20,250 shared origins) were reproducibly obtained in at least 20 independent SNS-seq datasets and contained in initiation zones identified by each of three other techniques, suggesting extensive variability in origin usage and identification. Also, 21% of the shared origins overlap with transcriptional promoters, posing a conundrum. Although the shared origins overlap more than union origins with constitutive CTCF-binding sites, G-quadruplex sites, and activating histone marks, these overlaps are comparable or less than that of known transcription start sites, so that these features could be enriched in origins because of the overlap of origins with epigenetically open, promoter-like sequences. Only 6.4% of the 20,250 shared origins were within 1 kb from any of the ~13,000 reproducible ORC-binding sites in human cancer cells, and only 4.5% were within 1 kb of the ~11,000 union MCM2-7-binding sites in contrast to the nearly 100% overlap in the two comparisons in the yeast, *Saccharomyces cerevisiae*. Thus, in human cancer cell lines, replication origins appear to be specified by highly variable stochastic events dependent on the high epigenetic accessibility around promoters, without extensive overlap between the most reproducible origins and currently known ORC- or MCM-binding sites.

## eLife assessment

The article addresses the mechanism of initiation of DNA replication in human cells by analyzing published data on the location of origins of DNA replication and the location of binding sites in the genome for ORC and MCM2-7 complexes. There are some **useful** analyses of existing data but there are concerns regarding the conclusion that there might be alternative mechanisms for determining the location of origins of DNA replication in human cells compared to the well-known mechanism known from many eukaryotic systems, including yeast, *Xenopus*, *C. elegans,* and *Drosophila*. The lack of overlap between binding sites for ORC1 and ORC2, which are known to form a complex in human cells, is a particular concern and points to the evidence for the accurate localization of their binding sites in the genome being **incomplete**.

## Introduction

DNA replication is essential for the duplication of a cell and the maintenance of the eukaryotic genome. To replicate the human diploid genome of~3billion base pairs, efficient cellular programs are coordinated to ensure genetic information is accurately copied. In each human cell cycle, replication starts from~50,000 genomic locations called replication origins (*Hu and Stillman, 2023*). At an origin of replication, the double-stranded DNA is unwound, and primase-DNA polymerase alpha lays down the first RNA primer extended as DNA. Once replication is initiated, the helicase involved in unwinding the origin continues to unwind DNA on either side and the replication proteins, including DNA polymerases, help copy the single-stranded DNA. Unlike yeast origins, human and most eukaryotic origins do not seem to have a clear DNA sequence preference (*Hu and Stillman, 2023*; *Leonard and Méchali, 2013*). Yet it becomes clear that chromatin context and other DNA-based activity are important factors for human and non-yeast eukaryotic origin selection. A great amount of effort has been spent to understand how human origins are specified and sometimes different trends are observed regarding genomic features enriched at human origins. For example, G-quadruplex motifs were found to be associated with human origins (*Besnard et al., 2012*), but can only explain a small subset of origins even after controlling for technical biases (*Foulk et al., 2015*). To better resolve these issues, we reasoned that a systematic analysis of all available human origin-mapping datasets will help uncover potential technical and biological details that affect origin detection.

To profile replication origins genome-wide, several sequencing-based methods have been developed, including short nascent strand (SNS)-seq (*Foulk et al., 2015*), Repli-seq (*Hansen et al., 2010*), Rerep-seq (*Menzel et al., 2020*), and Bubble-seq (*Mesner et al., 2013*). SNS-seq assays across different human cell types identified a subset of origins that can explain ~80% of origins in any tested cell type (*Akerman et al., 2020*); however, no study has systematically examined the consistency within and between different origin-mapping techniques. These methods are based on different molecular capture strategies and therefore are expected to have technical biases. For instance, SNS-seq utilizes lambda exonuclease ($\lambda$-exo) to enrich RNA-primed newly replicated DNA by removing parental DNA fragments. However, the products may be contaminated with chromosomal DNA fragments or be affected by the cutting bias of $\lambda$-exo against guanine-cytosine (GC)-rich sequences (*Foulk et al., 2015*), although it has been proposed that one can experimentally correct and control for such biases (*Akerman et al., 2020*). SNS-seq is the only method that yields origins at a resolution of a few hundred base pairs, with all other methods delineating larger initiation zones (IZs) that may have multiple origins. Repli-seq relies on nucleotide pulse labeling and antibody enrichment of newly replicated DNA superimposed on cells fractionated at different parts of the S phase by flow cytometry. Here the results may be contaminated by DNA non-specifically associated with the antibody (*Zhao et al., 2020*). Rerep-seq only captures the new synthesized sequences that replicate more than once when replication is dysregulated, and these origins tend to be enriched in the early replicating, epigenetically open parts of the genome (*Menzel et al., 2020*). Bubble-seq (*Mesner et al., 2013*) generates long reads because it captures the origin-containing replication intermediates by selection of DNA bubbles, and so may enrich origins that are flanked by pause sites. Okazaki-seq (OK-seq) identifies sites where the direction of the Okazaki fragments changes from leftward to rightward on the chromosome-revealing sites where the two lagging strands diverge from each other (*Petryk et al., 2016*). Because the technical biases mentioned above are associated with specific techniques, we suspect that origins captured across different techniques are less likely to be affected by those biases. Given the large number of datasets in the public domain (>100 datasets for human origins), this is an opportune time to study the reproducibility between the studies and determine the most reproducible and consistent origins identified by SNS-seq and confirmed by each of the other methods. This most reproducible group of origins (shared origins) from different research groups using different techniques and different cell lines will be far fewer than all the origins reported to date and minimize complications from stochastic (noisy) firing of origins, but they are the best expected to fit the current hypotheses regarding origin specification.

Yeast origin recognition complex (ORC) binds to double-stranded DNA with sequence specificity, helps load minichromosome maintenance protein complex (MCM), and thus prepares the origins for subsequent firing (*Bell and Dutta, 2002*; *Remus et al., 2009*; *Bell and Stillman, 1992*; *Costa and Diffley, 2022*). Consistent with this, the role of ORC in loading MCM proteins was also described in *Xenopus* egg extracts (*Rowles et al., 1999*; *Coleman et al., 1996*). All of this leads to the expectation

that in human cells there will be significant concordance between ORC-binding sites and efficient origins of replication. Efforts to define double-stranded DNA sequences bound specifically by human ORC find very little sequence specificity (*Vashee et al., 2003*; *Hoshina et al., 2013*), and this may be responsible for the lack of sequence specificity in human origins. This difference between yeast and human ORC is attributed to sequence features in ORC4, one of the subunits of the ORC (*Lee et al., 2021*). When a yeast-specific 19 amino acid insert was removed from yeast ORC4 to make it more like the human ORC4, the sequence specificity of ORC binding was lost even in yeast. Despite this loss of sequence specificity, the mutant yeast ORC loaded MCM proteins, and the yeast replicated DNA and survived. Thus, even if human origins of replication do not have specific sequences, one should expect some concordance between ORC-binding sites and the most reproducible origins of replication. An additional complexity is that genome-wide analysis found active origins and dormant origins determined by SNS-seq and OK-seq have little difference in ORC or MCM density (*Sugimoto et al., 2018*; *Kirstein et al., 2021*). Finally, a few publications have reported significant MCM2-7 loading and DNA synthesis after genetic mutation of *Drosophila ORC1* and human *ORC1*, *ORC2*, or *ORC5* genes that reduce the expressed proteins to near undetectable levels (*Park and Asano, 2008*; *Shibata et al., 2016*; *Okano-Uchida et al., 2018*; *Shibata and Dutta, 2020*). In two of the instances, the DNA replication seen was in the context of endo-reduplication, a process that is still believed to require the loading of MCM2-7 and the activation of the same into an active CMG helicase. It should be noted, however, that sgRNA screens revealed that the *ORC2* gene was essential for viability in the cancer cell lines mutated for *ORC2*, suggesting that vanishingly small amounts of the *ORC2* gene product are required for some processes essential for cell proliferation (*Chou et al., 2021*). Definition of the most reproducible and consistent origins identified by different methods from different research groups in different cell lines thus provide a unique opportunity to determine how much overlap is seen between the highly reproducible human origins and the ORC-binding sites reported in the literature from ChIP-seq studies.

To understand the genome-wide distribution patterns of replication origins in an unbiased way, we performed an integrative analysis of 113 DNA replication origin datasets. Because SNS-seq is unique in identifying origins of a few hundred bases long, while the other methods identify larger IZs that contain origins, we first prepared a list of high-resolution SNS-seq origins that have been identified in at least 20 of the 66 SNS-seq datasets in this study. We next determined those reproducible SNS-seq origins that overlap with IZs identified by each of the three techniques (Repli-seq, OK-seq, and Bubble-seq) to identify 20,250 high-confidence origins that are shared between SNS-seq and every other method identifying IZs. Using these shared origins, which overlap significantly with transcription promoters, we tested whether G-quadruplex sites, CTCF-binding sites, ORC-binding sites, or MCM-binding sites help specify origins.

## Results

### A total of 7,459,709 origins from 113 datasets show similar but different genomic features associated with each origin-mapping technique

We collected 113 publicly available replication origin identification datasets in different human cell types from five different techniques (*Figure 1—figure supplement 1a*), including SNS-seq, Repli-seq, Rerep-seq, OK-seq, and Bubble-seq. The complete list of datasets used in this analysis can be found in *Supplementary file 1*. We processed all the datasets using the first two steps of a pipeline with different parameters considering the various resolutions of different techniques as SNS-seq is unique in identifying high-resolution origins of replication (*Figure 1a*). Each of the 113 datasets yielded at least 1000 origins. We merged origins that overlap for at least 1bp from each other and cut the merged regions into 300bp segments, considering origin lengths were significantly longer for Bubble-seq and OK-seq, methods known to identify IZs (*Figure 1—figure supplement 1b*). A total of ~7,460,000 union origins were discovered from all techniques (*Figure 1b*).

Principal component analysis (PCA) of all origin datasets shows that origin profiles from the same technique are more similar to each other than from different techniques (*Figure 1c*). This is confirmed by pairwise correlations of the datasets, where it is evident that each method identifies origins that are best correlated with origins identified by that method alone (*Figure 1—figure*

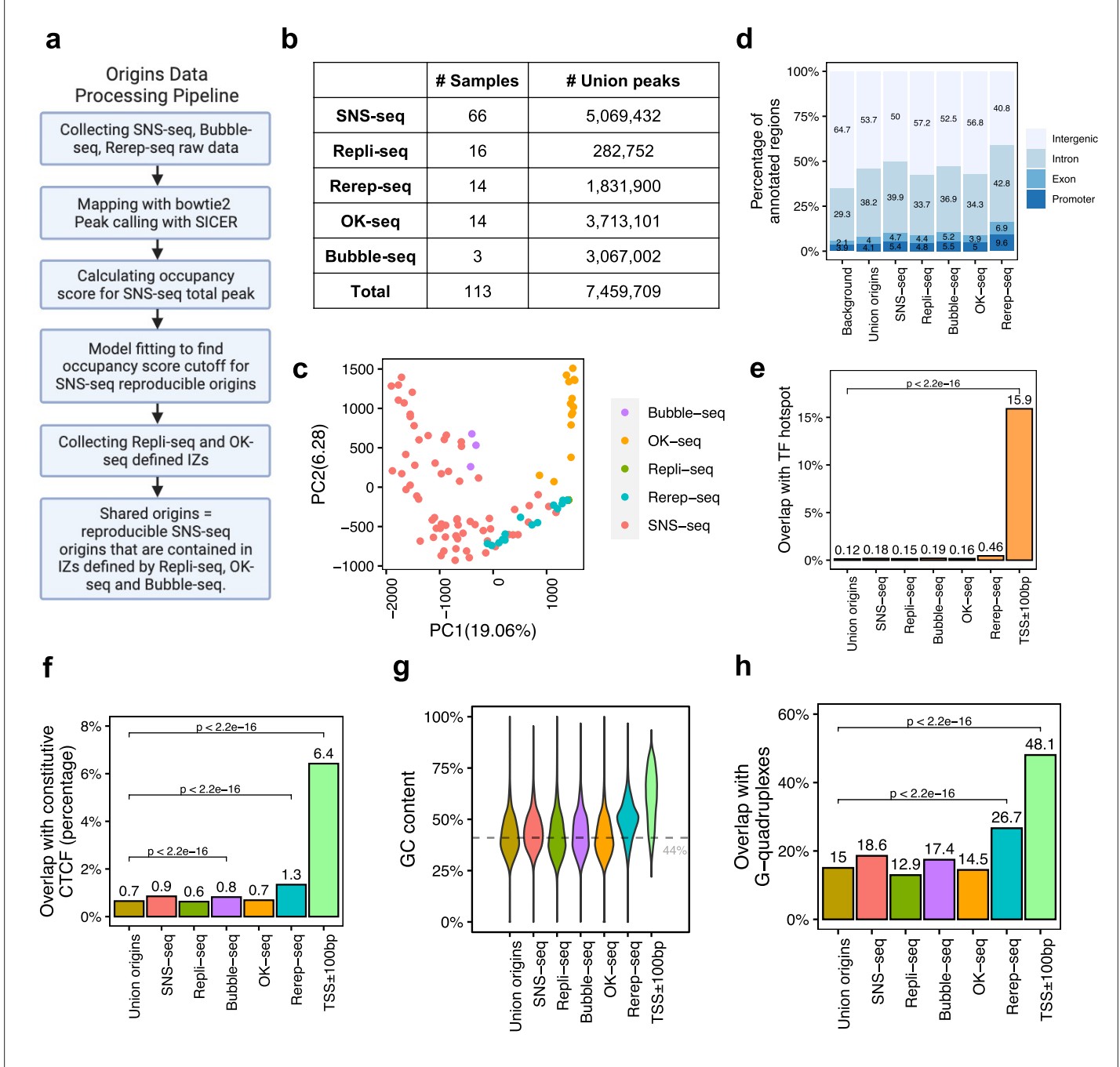

**Figure 1.** A total of 7,459,709 origins defined by four types of techniques show different genomic features. (**a**) Data processing pipeline. A total of 113 publicly available profiles of origins are processed following the pipeline. (**b**) Number of samples collected for each technique. In total, 7,459,709 union origins were identified. (**c**) Principal component analysis (PCA) shows the clustering of origin datasets from different techniques. (**d**) Genomic annotation (transcription start sites [TSS], exon, intron, and intergenic regions) of different groups of origins. Background is the percentage of each annotation on the whole genome. (**e**) Overlap with TF hotspots for different groups of origins and promoters. (**f**) Overlap with constitutive CTCF-binding sites for different groups of origins and promoters. (**g**) GC content of different groups of origins and promoters. Gray line marks the average GC content of the human genome. (**h**) G-quadruplex overlapping rates of different groups of origins and promoters.

The online version of this article includes the following figure supplement(s) for figure 1:

**Figure supplement 1.** Distribution of origins defined by four types of techniques.

**Figure supplement 2.** Correlation between origins from different samples.

supplement 2). Although the most popular technique, SNS-seq, shows some internal variability in the PCA (*Figure 1c*), it is also the one that is best correlated with origins identified by other SNS-seq datasets. Rerep-seq and Bubble-seq seem to identify IZs that are also better correlated with SNS-seq origins (*Figure 1—figure supplement 2*). Since Rerep-seq tends to identify the origins that are enriched in early replicating, gene-rich parts of the genome, this result suggests that SNS-seq and Bubble-seq also have some bias for origins in early replicating, gene-rich parts of the genome. However, we did not observe clear trends of clustering of origins identified in similar cell types (*Figure 1—figure supplement 1c*), even when we focused only on SNS-seq origins (*Figure 1—figure supplement 1d*). The one exception was the SNS-seq origins from T lymphoblasts, which were very closely clustered with each other in six datasets, but these were all done in the CCRF-CEM cell line (or a derivative) by one group at the same time (*Murai et al., 2018*). We only examined SNS-seq data from 2018 onward when the lambda exonuclease-based digestion step had been incorporated into SNS-seq protocols, but wondered whether there was a steady improvement of reproducibility as one progressed through the years. However, SNS-seq datasets do not become progressively more reproducible as one goes year-by-year from 2018 to 2022 (*Figure 1—figure supplement 1e*). The results suggest that there is a significant difference in origins captured by different techniques that cannot be explained by differences in cell choices. For the most popular technique (SNS-seq), the variability may be least when confined to a single cell line studied at the same time by the same group.

We next checked the genomic characteristics of origins defined by each technique. Regardless of the technique used, around half of the detected origins fall in intergenic regions, followed by 30–40% allocated to introns (*Figure 1d*). We recently reported 40,110 genomic regions are routinely bound by more than 3000 transcription factor (TF) ChIP-seq data, named as TF-binding hotspots (*Hu et al., 2021*). These TF-binding hotspots likely represent highly active open chromatin regions. Despite the differences among techniques, the origins in general have lower overlap with TF-binding hotspot (<0.5%) compared to the 16% overlap of gene promoter regions (transcription start sites [TSS]± 100bp) with such TF-binding hotspots (*Figure 1e*). We further focused on one specific TF, CCCTC-binding factor (CTCF), which is important in organizing DNA structure and reported to be associated with replication (*Su et al., 2020*). Origins show significantly lower overlap with constitutive CTCF sites, defined as those that are conserved across cell types (*Fang et al., 2020*), compared to promoters (*Figure 1f*). G-quadruplexes are found to be correlated with both replication and transcription (*Lipps and Rhodes, 2009*). However, compared to promoters, origins defined by these techniques show significantly lower GC content (*Figure 1g*). In addition, there is a significantly lower overlap of these origins with G-quadruplex sites identified from the predicted G-quadruplex motif regions (*Bedrat et al., 2016*) compared to the overlap of promoters with G-quadruplex sites (*Figure 1h*). Thus, the reported correlations between CTCF-binding sites or G-quadruplexes with origins are not as striking as that of these features with promoters.

We found that all origins, regardless of the technique by which they are identified, are slightly enriched at gene promoters, exons, or introns compared with the genome background (*Figure 1d*). Among the five different techniques, origins defined by Rerep-seq show the highest level of enrichment with promoters, TF hotspots, constitutive CTCF-binding sites and G-quadruplexes, and the highest GC content. This is consistent with our knowledge that areas of the genome that are re-replicated when the cell cycle is disturbed are enriched in parts of the genome that replicate early in the S phase, regions that are enriched in transcriptionally active (and thus epigenetically open) genes and their promoters.

## Shared origins are associated with active chromatin and transcription regulatory elements

To address the potential biases caused by each technique, we investigated how many of the ~5 million union SNS-seq origins are reproducible in SNS-seq data and confirmed by other sequencing-based techniques. As we will describe below, these shared origins show significant overlap with promoters. Because the Rerep-seq origins appear to be slightly different from the origins identified by other techniques, with the highest overlap with transcriptionally active genes and promoters, we decided to exclude the Rerep-seq from the analysis of shared origins and still reached the conclusion summarized above.

SNS-seq origins have the highest resolution. We used the following strategy to determine how many independent confirmations of an SNS-seq origin are sufficient for selecting an SNS-seq origin as a reproducible origin. The occupancy score of each origin defined by SNS-seq (*Figure 2—figure supplement 1a*) counts the frequency at which a given origin is detected in the datasets under consideration. Plotting the number of union SNS-seq origins with various occupancy scores with all SNS-seq datasets published after 2018, we sought to determine whether the curve deviates from the random background at a given occupancy score (*Figure 2a*). For the random background, we assumed that the number of origins confirmed by increasing occupancy scores decreases exponentially (see 'Methods' and *Supplementary file 2*). The threshold occupancy score, to determine whether an origin is a reproducible origin, is the point where the observed number of origins deviates from the expected background number (with an empirical FDR < 0.1) (*Figure 2a*), an occupancy score of 20. Thus, the reproducible SNS-seq origins (with an FDR < 0.1) were those observed in at least 20 SNS-seq datasets.

We next determined which reproducible SNS-seq origins are confirmed by origins from Repli-seq and replication IZs from Bubble-seq and OK-seq (*Figure 2b*). A total of 20,250 of the reproducible SNS-seq origins were found to overlap with an origin or IZ identified by each of the three other techniques and were called shared origins. These high-confidence shared origins consist of 0.27% of all ~7.5 million union origins. The coordinates of the shared origins are available in the supplementary files.

The shared origins have a greater overlap rate with gene promoters and exons compared to union origins (*Figure 2c*). This is in line with the previous observation that replication and transcription are highly coordinated and enrichment of origins at TSS (*Ganier et al., 2019*; *Cook, 1999*; *Karnani et al., 2010*). Shared origins also have a substantially higher overlap rate than union origins with TF-binding hotspots (*Figure 2d*). Moreover, shared origins have a higher overlap rate with constitutive CTCF-binding sites compared to union origins (*Figure 2e*) and have a higher GC content and overlap with G-quadruplexes than union origins (*Figure 2f and g*).

BART (*Wang et al., 2018*) analyzes the enrichment of TF-binding sites (determined experimentally by ChIP-seq) with areas of interest in the genome. We used BART to perform TF-binding site enrichment analysis on the shared origins and union origins and identified potential TFs or components of chromatin remodeling factors (e.g., PAF1, EZH2, ZNF282, POLR2A, GTF2I) whose binding sites are associated with the shared origins (*Figure 2h*, *Figure 1—figure supplement 1f*). The high enrichment of activators or repressors of transcription in the factors that have binding sites near the shared origins provides more support that the shared origins have properties similar to transcriptional promoters.

To investigate whether the shared origins have a specific chromatin epigenomic signature compared to union origins, we used 5711 publicly available ChIP-seq datasets for 29 different histone modifications and generated a comprehensive map of histone modification enrichment at shared origins compared to union origins. A substantial enrichment of activating histone marks, including H3K4me3 and H3/H4 acetylation, was observed at shared origins compared to union origins (*Figure 2i*). The enrichment of H3K14ac is interesting given the enrichment in the BART analysis of binding sites of BRPF3, a protein involved in this specific acetylation and reported to stimulate DNA replication (*Feng et al., 2016*), but this modification was not uniquely enriched at the shared origins. These results show that as we move from all origins to a small set of high-confidence shared origins, we see a progressive increase in the enrichment of TSS in epigenetically active parts of the genome.

However, transcriptional and epigenetic activators are not the whole story. H3K27me3, a repressive mark, is also enriched at shared origins, although this enrichment is not as high as that of the activating marks. The enrichment of EZH2-binding sites near the shared origins is consistent with this observation because EZH2 is part of the Polycomb Repressive Complex 2, known to be a writer of H3K27me3. The paradoxical enrichment of repressive marks is consistent with the shared origins also being near the binding sites of MBD (binds methylated DNA and represses transcription), PATZ1 and ZNF282, proteins known to be repressors of transcription.

Having identified a small group of highly reproducible origins (independent of cell type and technique), we asked whether in a given cell line was one technique superior to the others in identifying such origins. K562 cells have been interrogated for origins by three different techniques: SNS-seq, OK-seq, and Repli-seq (*Figure 2—figure supplement 2*). Among the 20,250 high-confidence shared origins, 9901 (48.9%) overlapped with SNS-seq origins in K562, 3872 (19.1%) overlapped with OK-seq IZs in K562, and 1163 (5.7%) overlapped with Repli-seq origins in K562. This suggests that in one cell

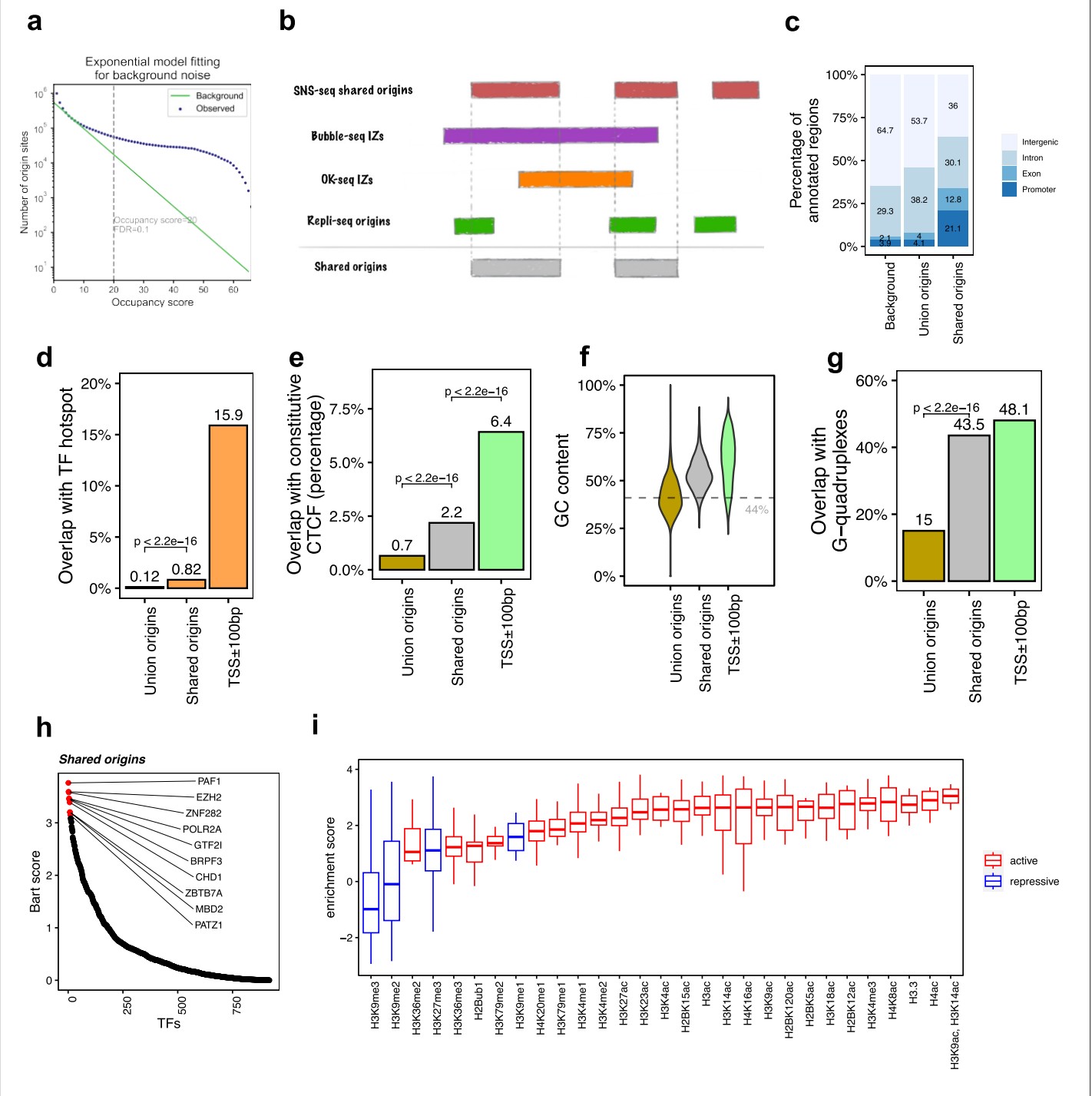

**Figure 2.** The shared origins are enriched with certain transcription factors and active histone marks. (**a**) Short nascent strand-seq (SNS-seq) origin-fitting distribution to an exponential model shows an occupancy score ≥20 is selected for reproducible SNS-seq origins. (**b**) Conceptual model of how the shared origins are determined. Any SNS-seq shared origin that overlaps with Bubble-seq initiation zone (IZ), Okazaki-seq (OK-seq) IZ, and Repli-seq origin together is considered as an origin identified by all four techniques (shared origins). (**c**) Genomic annotation of union origins and shared origins. (**d**) Overlap with TF hotspots of union origins and shared origins. (**e**) Overlap with constitutive CTCF-binding sites of union origins and shared origins. (**f**) GC content of union origins and shared origins. (**g**) G-quadruplex overlapping rates of union origins and shared origins. (**h**) BART prediction of TFs associated with shared origins. (**i**) Enrichment of histone marks at shared origins using all union origins as control.

The online version of this article includes the following figure supplement(s) for figure 2:

**Figure supplement 1.** Background model for the identification of the shared origins.

*Figure 2 continued on next page*

*Figure 2 continued*

**Figure supplement 2.** Origins/IZs defined by different techniques: (**a**) SNS-seq, (**b**) OK-seq and (**c**) Repli-seq in K562 cell line and their overlap with the shared origins.

line even the large IZs defined by OK-seq or Repli-seq do not capture most of the high-confidence origins. In the opposite direction, where we estimate what fraction of origins found by a given technique fall in a reproducible origin, the opposite result emerges: 2.7% of SNS-seq origins, 17.2% of OK-seq IZs, and 7.7% of Repli-seq IZs overlapped with the 20,250 shared origins (*Figure 2—figure supplement 2*). Thus, SNS-seq may be able to identify more of the reproducible origins, but it comes with a high false-positive rate.

## Human, but not yeast, high-confidence origins have low overlap with known ORC-binding sites

ORC is expected to bind near replication origins during the cell cycle to help define origins (*Bell, 2002*). To investigate the correlation of ORC binding with shared origins detected across different origin-mapping techniques, we analyzed all five publicly available ORC1 and ORC2 ChIP-seq datasets with at least 1000 peaks (*Supplementary file 1c*). We identified a total of 34,894 ORC-binding sites in the human genome (*Supplementary file 3*). Union (all) of ORC-binding sites are enriched at promoters, TF hotspots, constitutive CTCF sites, GC content, and G-quadruplexes compared to randomized genome background (*Figure 3a–e*), somewhat similar to what we observed for the shared origins (*Figure 2*).

Of the 34,894 ORC-binding sites in the human genome, 12,712 sites were defined as shared ORC-binding sites as they occur in at least two ORC datasets (occupancy score ≥ 2). For the majority of genomic features, including enrichment at the promoter (*Figure 3a*), overlap with constitutive CTCF sites (*Figure 3c*), GC content (*Figure 3d*), and overlap with G-quadruplex (*Figure 3e*), ORC-binding sites have similar genomic distribution regardless of how many samples they appear in. Interestingly, the G-quadruplex enrichment at either union or shared ORC-binding sites is a bit lower than that in gene promoter regions (*Figure 3e*). The only significant difference between the union ORC sites and the shared ORC-binding sites is the significantly higher overlap of the latter with TF-binding hotspots (*Figure 3b*), suggesting that ORC-ChIP seq data also tend to enrich highly open chromatin regions.

Based on the assumption that ORC-binding sites that appeared in more than one dataset are likely to be true-positive ORC-binding sites, we analyzed how many of the 12,712 shared ORC-binding sites *overlap* with the 20,250 shared origins identified by our analysis shown above. The overlap was surprisingly low: 1.7% of the shared origins overlapped with shared ORC-binding sites (*Figure 3f*, *Figure 3—figure supplement 1a*). Even when we relaxed the criteria to look at the shared origins that are proximal (≤1 kb) to the shared ORC-binding sites, only 1300 (6.4%) of the shared origins were proximal to the shared ORC-binding sites (*Figure 3f*; *Supplementary file 4b*). The low degree of overlap or proximity of shared origins with shared ORC-binding sites indicates that the vast majority of the shared origins are not near the experimentally determined ORC-binding sites.

In the reverse direction, we asked whether an ORC-binding site definitively predicts the presence of an origin nearby. A histogram of the distance between any ORC-binding site (union) and the nearest shared origin showed that only 1086 (3.11%) ORC-binding sites are proximal to (≤1 kb) a shared replication origin (*Figure 3g*, *Supplementary file 4a*). Given that there are 34,894 ORC-binding sites and 20,250 shared origins, if ORC binding was sufficient to determine a high-confidence origin, nearly half of the ORC-binding sites should have been proximate to the shared origins. This low level of proximity between ORC binding and reproducible origins suggests that the current data on ORC-binding sites is unable to predict the presence of a high-confidence, actively fired origin.

We examined whether the co-localization is better if the analysis is done with data exclusively from the same human cell type (*Figure 3—figure supplement 1b and d*). Only 8.8% of the 105,881 union origins identified by SNS-seq, OK-seq, or Repli-seq in K562 cells overlap with ORC2-binding sites mapped in the same cell line, and only 4.9% of the 68,003 union SNS-seq origins mapped in HeLa cells overlap with ORC1-binding sites in the same cell line. The overlaps improve marginally if we focus on the shared origins: 12.8% of 9605 shared origins in K562 cells and 6.1% of 3390 shared origins in HeLa cells overlap with ORC2 and ORC1 ChIP-seq sites in the concordant cell lines.

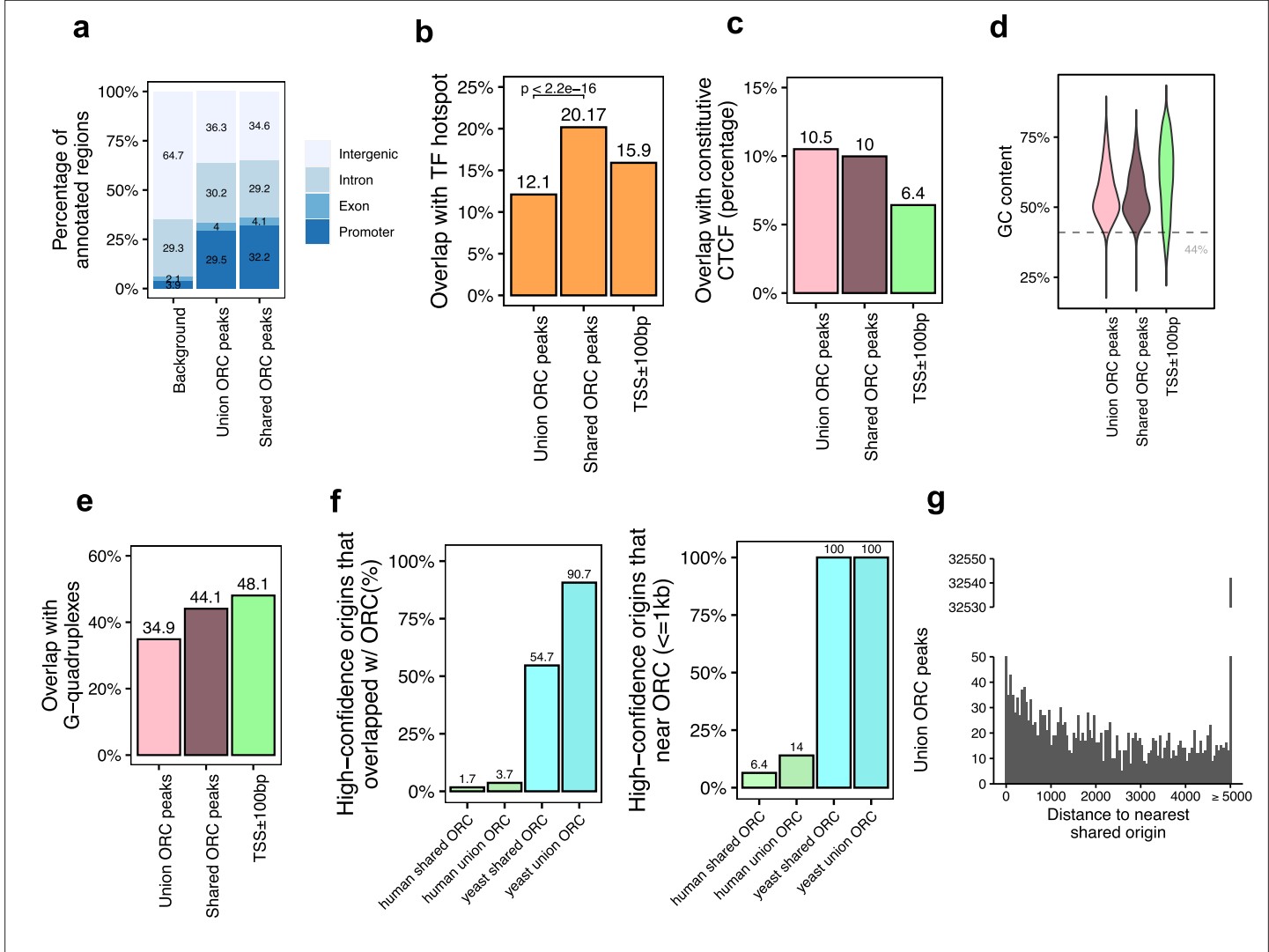

**Figure 3.** Genomic features of the shared origin recognition complex (ORC)-binding sites and their co-localization with the shared origins. (**a**) Genomic annotation of union ORC and shared ORC-binding sites. (**b**) Overlap with TF hotspot of union ORC and shared ORC-binding sites. (**c**) Overlap with constitutive CTCF-binding sites of union ORC and shared ORC-binding sites. (**d**) GC content of union ORC and shared ORC-binding sites. (**e**) Overlap with G-quadruplex of union ORC and shared ORC-binding sites. (**f**) The percentage of high-confidence origins (shared origins in humans and confirmed origins in yeast) that overlapped with (left) or are proximate to (≤1 kb) (right) two types of ORC-binding sites (union or shared). (**g**) Distribution of the distance between ORC-binding sites and the nearest shared origin.

The online version of this article includes the following figure supplement(s) for figure 3:

**Figure supplement 1.** Analysis of overlap between the shared origin recognition complex (ORC)-binding sites and origins.

Experiments in the yeast *Saccharomyces cerevisiae* have demonstrated that ORC binding is critical for defining the origin of replication. Indeed, in contrast to humans, in the more compact genome of the yeast, *S. cerevisiae*, there is higher overlap or proximity (within 1 kb) seen between the yeast ORC ChIP-seq-binding sites (*Supplementary file 1*) and the well-mapped yeast origins of replication in OriDB (*Nieduszynski et al., 2007*). In this case, 100% of the shared origins are proximate to (≤1 kb) all (union) or shared ORC-binding sites in yeast (*Figure 3f*).

Overall, we found that the shared origins are highly co-localized with ORC-binding sites in yeast but not in the human cell lines, suggesting that the current methods of ORC-binding site determination are failing to identify functional ORC binding.

## Properties of the 6.4% shared origins co-localized reproduced ORC-binding sites

Because of the poor co-localization of shared origins with ORC-binding sites, we next focused on 6.4% of shared origins (1300) that are proximate to a shared ORC-binding site, a group we will call the 'highest-confidence origins'. Compared with the shared origins or ORC-binding sites alone, these 1300 origins are even more co-localized with gene promoters (*Figure 4a*), TF-binding hotspots (*Figure 4b*), and constitutive CTCF-binding sites (*Figure 4c*). The GC content is not significantly higher for these 'highest confidence origins' (*Figure 4d*), and neither was their overlap with G-quadruplexes (*Figure 4e*).

To understand the correlation between the different classes of origins and replication timing, we calculated the replication timing score for the origins following the ENCODE pipeline (*Navarro Gonzalez et al., 2021*; *Hansen et al., 2010*) and found that the mean replication timing score of shared origins suggests that they replicate earlier in the S phase compared to the union origins, but there is not much difference between the shared origins separated by whether they overlapped with the ORC-binding sites or not (*Figure 4f*).

This suggests that even though the shared origins near the ORC-binding sites are more enriched in TSS, TF-binding hotspots, and CTCF-binding sites they are still similar to the shared origins globally in their localization in early replicating, epigenetically active parts of the genome.

To investigate whether the 'highest confidence origins' (shared origins near the ORC-binding sites) are associated with higher transcriptional activity, we divided all human genes into four groups based on their expression level in the human K562 cell line and found that the genes co-localized with the highest confidence origins exhibit higher expression levels compared to the genes co-localized with all shared origins or union origins (*Figure 4g*).

BART analysis (*Wang et al., 2018*) of which protein-binding sites are most enriched in 743 origins overlapping with any ORC-binding sites shows, as expected, that the ORC2-binding sites are enriched near these origins. The other proteins bound near these origins are either transcriptional activators like EPC1, LEF1, ELK, EGR1, PAF1, or transcriptional repressors like KLF13, KRAB, ZNF639, or ZBTB33 (*Figure 4h*). These results suggest that the shared origins overlapping with the ORC-binding sites tend to be more associated with transcriptional regulation than all shared origins.

## Overlap of MCM-binding sites with the shared origins to define another type of the highest confidence origins

The six subunit minichromosome maintenance complex (MCM) is loaded on chromatin in G1 and forms the core of the active helicase that unwinds the DNA to initiate DNA replication (*Madine et al., 2000*). Since MCM2-7 may be loaded by ORC and move away from ORC to initiate DNA replication, it could be expected that even if the ORC-binding sites are not proximate to the 20,250 shared origins, they will be proximate with known MCM-binding sites. To test this, we analyzed 18 human MCM ChIP datasets (*ENCODE Project Consortium, 2012*; *Ivanov et al., 2018*; *Utani et al., 2017*). We identified a total of 11,394 MCM3-7-binding sites (union) and 3209 shared MCM-binding sites that are defined by an intersection of MCM3, MCM5, and MCM7 union peaks. Overall, MCM-binding sites displayed very similar genomic features to the ORC-binding sites (*Figure 5a–e*). We then defined the genomic features common among the shared origins that are close to (≤1 kb) the MCM-binding sites. Like the ORC-designated origins (*Figure 4b*), the MCM-designated origins showed higher overlap with TF-binding hotspots (*Figure 5f*). Very interestingly, similar to the high overlap between yeast origins and yeast ORC sites (*Figure 3f*), around 95% of yeast origins (*Lee et al., 2021*; *Gros et al., 2015*) are close to (≤1 kb) the union of all yeast MCM sites. In contrast, only ~4.5% of shared human origins are close to (≤1 kb) the union of experimentally defined human MCM-binding sites (*Figure 5g*).

We examined all three types of high-confidence origins (ORC designated, MCM3-7 designated and MCM2 designated) in a Venn diagram (*Figure 5h*) and identified 74 that were reproducibly identified by multiple methods (shared origins near ORC-binding sites, MCM3-7-binding sites, and MCM2-binding sites). The coordinates of these origins and their supporting data (ORC- and MCM-binding sites) are listed in *Supplementary file 5*, and the genome browser views for three of them are shown in *Figure 6* to indicate the relationship between the coordinates.

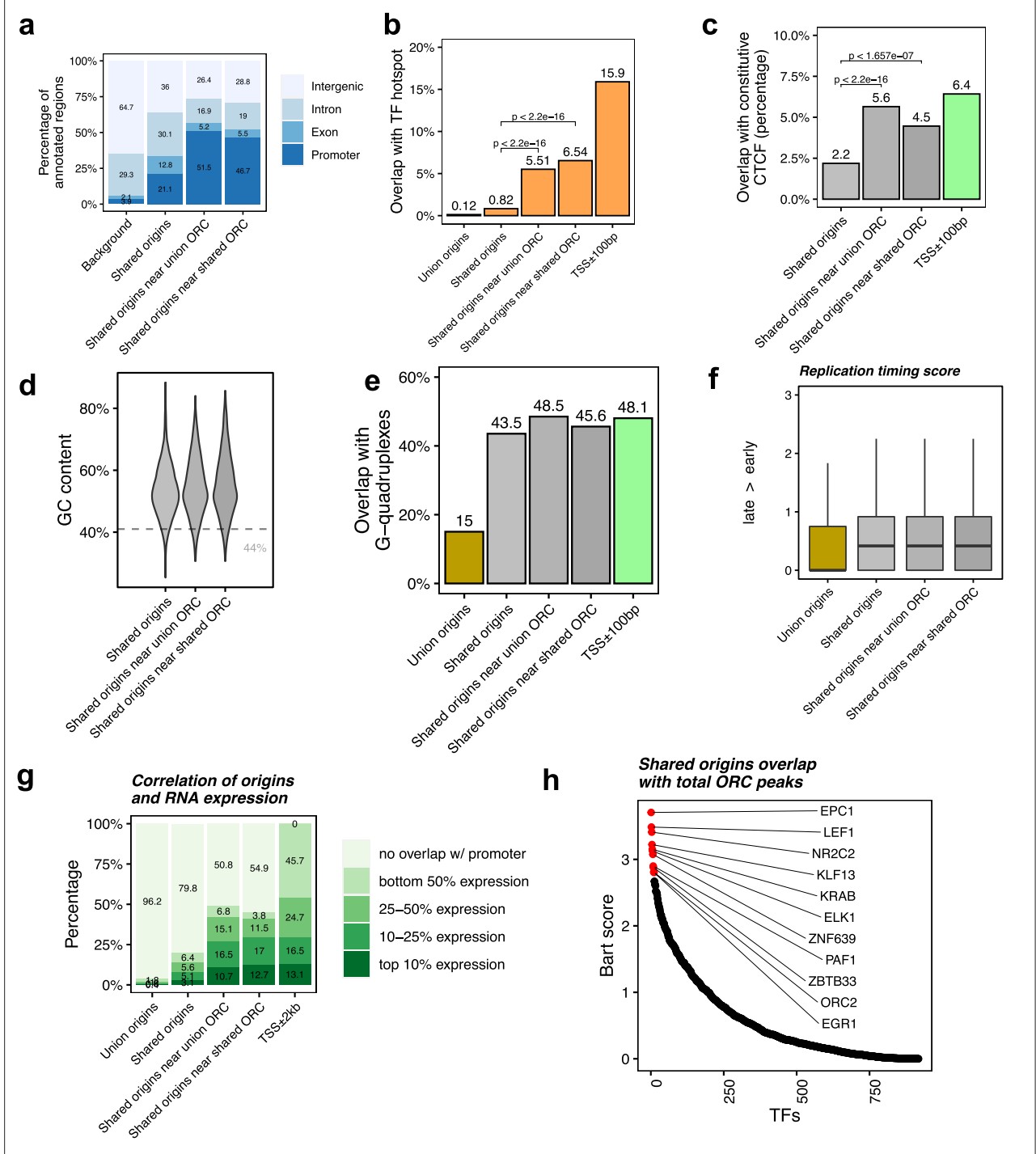

**Figure 4.** Shared origins near shared origin recognition complex (ORC)-binding sites are more correlated with active transcription. (**a**) Genomic annotation of shared origins and shared origins near (≤1 kb) the ORC-binding sites. (**b**) Overlap with TF hotspots of shared origins and shared origins near the ORC-binding sites. (**c**) Overlap with constitutive CTCF-binding sites of shared origins and shared origins near the ORC-binding sites. (**d**) GC content of shared origins and shared origins near the ORC-binding sites. (**e**) Overlap with G-quadruplex sites of shared origins and shared origins near the ORC-binding sites. (**f**) Y-axis: replication timing score from *Navarro Gonzalez et al., 2021* for the indicated classes of origins. (**g**) Annotation of expression level of genes that overlapped with different groups of origins. (**h**) BART prediction of TFs associated with the highest confidence origins.

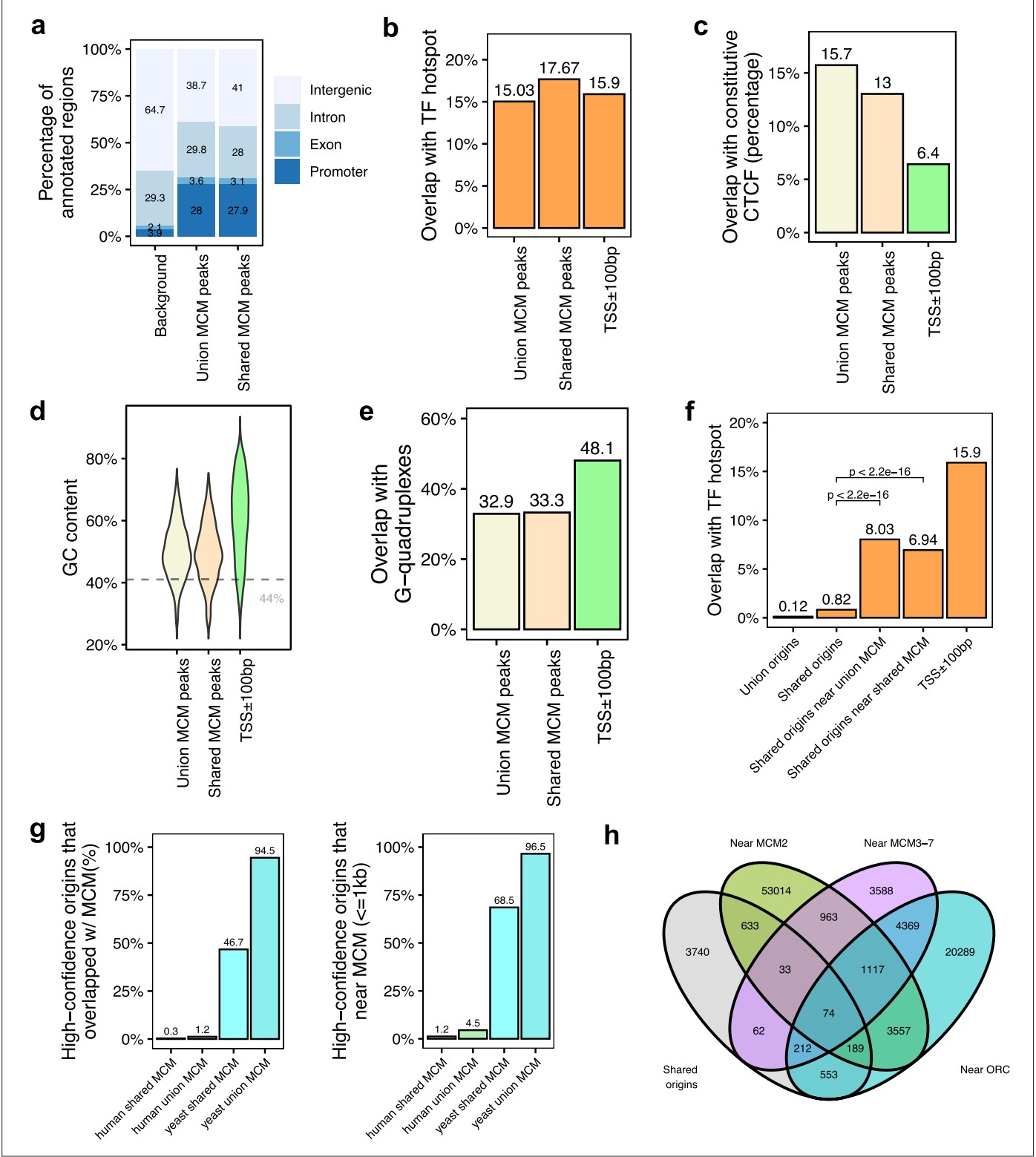

**Figure 5.** Genomic features of the shared minichromosome maintenance complex (MCM)-binding sites and their co-localization with the shared origins. (**a**) Genomic annotation of union MCM and shared MCM-binding sites. (**b**) Overlap with TF hotspot of union MCM and shared MCM-binding sites. (**c**) Overlap with constitutive CTCF-binding rates of union MCM and shared MCM-binding sites. (**d**) GC content of union MCM and shared MCM-binding sites. (**e**) Overlap with G-quadruplex of union MCM and shared MCM-binding sites. (**f**) Overlap with TF hotspots of shared origins and shared origins near the MCM-binding sites. (**g**) The percentage of high-confidence origins (shared origins in humans and confirmed origins in yeast) that overlapped

*Figure 5 continued on next page*

*Figure 5 continued*

with (left) or are proximate to (≤1 kb) (right) two types of MCM-binding sites (union or shared). (**h**) Venn diagram of the shared origins that are near the ORC-, MCM2-, or MCM3-7-binding sites.

The online version of this article includes the following figure supplement(s) for figure 5:

**Figure supplement 1.** Analyses of a few selected origin sets suggested by the reviewers.

**Figure supplement 2.** Origin recognition complex (ORC) subunits do not co-bind to DNA as much as expected.

**Figure supplement 3.** Shared origins overlap with phosphorylated MCM2.

**Figure supplement 4.** Selecting fewer but even more reproducible origins with more stringent cutoff to determine their overlap with the origin recognition complex (ORC) and minichromosome maintenance complex (MCM)-binding sites.

## Discussion

Through integrative analysis of publicly available 113 profiles of replication origins from multiple techniques, we identified ~7.5 million union origins in the human genome, of which only 20,250 shared origins are reproducibly identified by at least 20 SNS-seq datasets and confirmed by each of the three other techniques. The following conclusions were reached: the shared (highly reproducible) origins are only a small subset of all origins identified and are in epigenetically active chromatin with a strong preference for promoters. However, G-quadruplexes, CTCF-binding sites, and ORC-binding sites are also enriched near the promoters, so although these genomic features are enriched near the shared origins, it is not clear whether these genomic features actually specify origins or whether they are simply enriched near origins because the origins are near the promoters. Finally, our results suggest that the co-localization of the shared origins with ORC- or MCM2-7- binding sites is very low, much lower than that in yeast, and so (a) the currently known binding sites of these proteins should not be used as surrogate markers of origins in human cells and (b) there is a great need to improve our methods of identifying isoforms of these proteins that are strictly functional for origin firing. The current models of origin firing require ORC to bind near origins, to load MCM2-7 nearby, and the latter to be converted into active helicase to fire origins. To rigorously test this model, we need examples of reproducible origins where these conditions are met, and our study helps identify 74 such sites where such experimental verification can be carried out.

A careful analysis shows that individual methods produce origin datasets that are more correlated with each other than across methods. The extensive variation of origins between datasets is likely because of background noise in all current techniques and because there is extreme stochasticity of origin selection during DNA replication in individual cells and cell cycles in the same population of cells from a human cell line in culture. There are also differences that are created due to the differences in cell lineage. Despite this, we could cut through those differences to focus on the small subset of shared origins that seem to be active in multiple cell lineages.

The shared origins are enriched with active histone marks and enriched in early replicating parts of the genome that are overwhelmingly in an active epigenetic environment. H3K4me3 has been reported to be enriched at replication origins (*Picard et al., 2014*; *Cayrou et al., 2015*), and it has also been reported that demethylation of H3K4me3 by KDM5C/JARID1C promotes origin active (*Rondinelli et al., 2015*). H3ac/H4ac are also reported to be enriched at replication origins (*Cadoret et al., 2008*; *Sequeira-Mendes et al., 2009*), and this is regulated by various histone acetyltransferases and histone deacetylases (*Goren et al., 2008*; *Wang et al., 2009*). Interestingly, H3K27me3 has also been reported to be enriched at replication origins (*Picard et al., 2014*; *Cayrou et al., 2015*). The higher enrichment of *all* activating histone modifications relative to the repressive histone modifications (including H3K27me3) (*Figure 2j*) rather than individual types of modification suggests that the shared, highly reproducible origins are preferentially in epigenetically active euchromatin. This aligns well with the general enrichment of shared origins with TSSs and TF hotspots – which are concentrated in gene-dense, epigenetically open parts of the chromosomes. This also aligns well with the fact that the shared origins were not enriched for H3K9me3. H3K9me3 has been reported to be enriched near late replicating origins (*Wang et al., 2017*), while the shared origins are biased toward those that are in euchromatin. That origins, like TSS, prefer areas of the chromatin marked by activating epigenetic marks has been reviewed in 2016 (*Prioleau and MacAlpine, 2016*). We arrive at the same conclusion even though we worked with the origins identified after this review, presumably with significant improvements in method and even though we focused on the most reproducible origins.

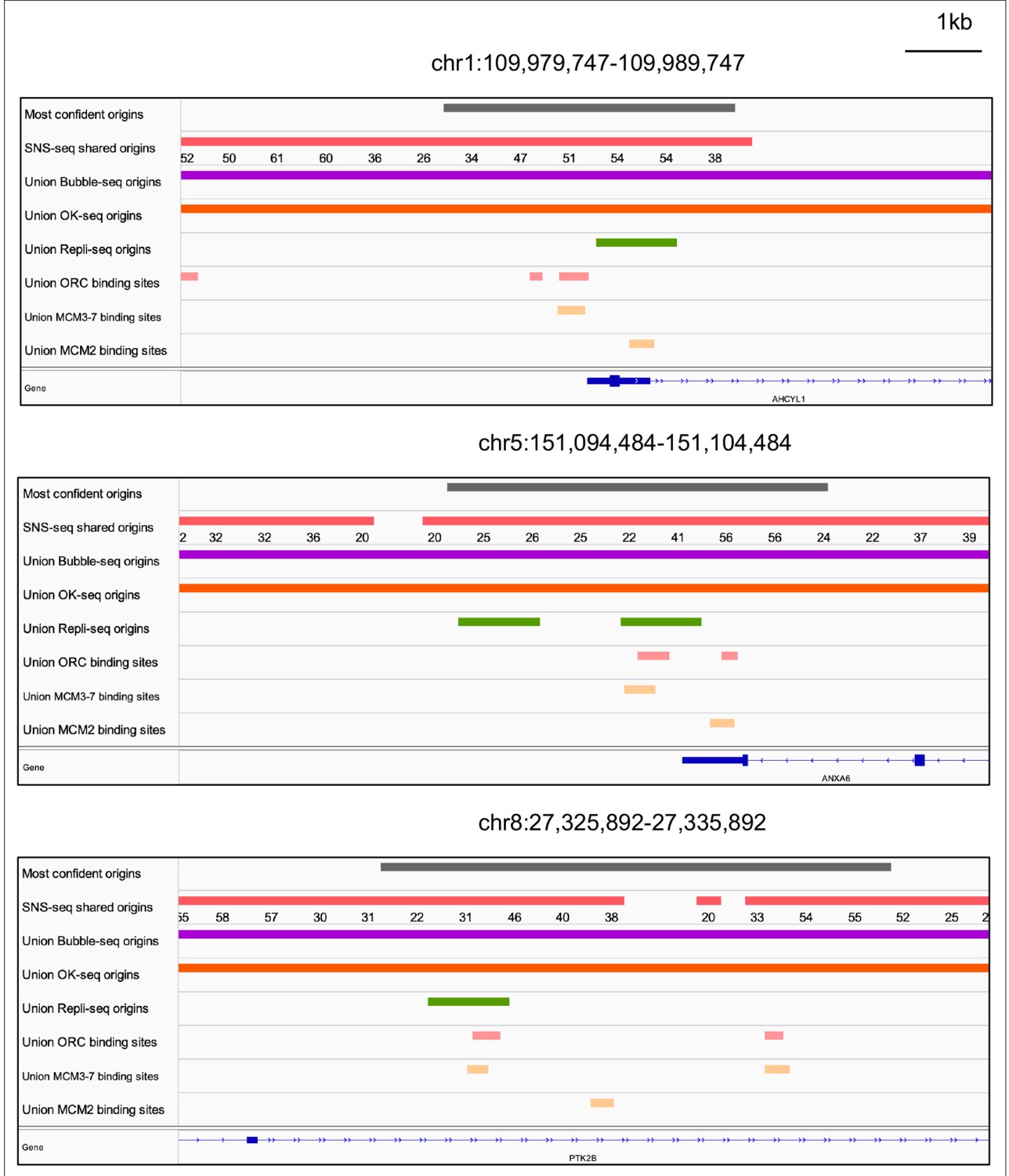

**Figure 6.** Genome browser screenshots for 3 of the 74 origins from *Figure 5h*. The numbers below the short nascent strand-seq (SNS-seq) shared origins track are the occupancy score of the origins along the length of the indicated track.

We characterized the genomic features of origins and found that the shared origins are more co-localized with TSS. *Table 1* shows a summary of the overlap of union origins, shared origins, and shared origins overlapping ORC-binding sites with various genomic features and compares this with the overlap of TSS with the same genomic features. Overall, these results suggest that as we proceed

**Table 1.** Overlap of origins, transcription start sites (TSS), and origin recognition complex (ORC)-binding sites with the indicated features.
Summary of data to show the number of origins of different types, extent of overlap of each with different genomic features, and comparison with TSS (or promoters).

| | Number | Promoter | TF hotspot | CTCF | GC content | G quad |
|---|---|---|---|---|---|---|
| | | | **% overlap with** | | | |
| Union origins | 7,459,709 | 4.1 | 0.1 | 0.7 | 41.3333 | 15 |
| Shared origins | 20,250 | 21.1 | 0.82 | 2.2 | 53 | 43.5 |
| Shared origins with shared ORC | 1300 | 46.7 | 6.54 | 4.5 | 54 | 45.6 |
| TSS ±-100 | 26,237 | 100 | 15.9 | 6.4 | 61.5 | 48.1 |
| Shared ORC | 12,712 | 32.2 | 20.7 | 10 | 53.1 | 44.1 |

from all origins to shared origins to the highest confidence origins that are proximate to the ORC-binding sites, we see increasing overlap with promoters and early replicating, transcriptionally active, epigenetically open parts of the genome and that features thought to be important for origin selection (like G-quadruplexes, CTCF-binding sites, active chromatin, ORC-binding sites) could simply co-occur with origins because of their enrichment near the promoters (TSS).

To rigorously test whether the overlaps of shared origins that we observe with various genomic features are significantly above the background, we performed a permutation test that evaluates whether the observed overlap is significantly above the mean expected overlap when the experimental dataset is randomized 1000 times (*Table 2*). The overlap of *shared origins* with promoters (TSS ±4 kb), G-quadruplexes, R-loops, and CTCF-binding sites are all significantly above the background. However, the overlap of *promoters* with G-quadruplexes, R loops, and CTCF-binding sites are also all significantly above the background, though the enrichment of G-quadruplexes is minimal. Thus, the genomic features characterized are all significantly enriched in both origins and promoters, and may help specify both, either independently or co-dependently.

G-quadruplexes are of particular interest because they have been experimentally shown to dictate origin specification (*Prorok et al., 2019*; *Valton et al., 2014*). Although there is a higher overlap of shared origins than union origins with G-quadruplexes (43.5% vs 15%), this overlap is not higher than that between promoters and G-quadruplexes (48.1%) (*Figure 2g*). This may suggest that the

**Table 2.** Permutation test of overlap of shared origins or promoters (transcription start sites) with the region around promoters, shared origin recognition complex peaks (in more than two datasets), R-loops, G-quadruplexes, and CTCF-binding sites.
Fold enrichment of the observed overlap relative to the mean overlap seen with 1000 randomizations of set A is indicated together with the p-values of the enrichment. Permutation test to ascertain the significance of the overlaps reported in this article relative to random expectation.

| Set_A | Set_B | #Set_A | #Set_B | Observed overlap | # Random iterations | Mean random overlap | Obs/ random fold enrichment | p-Value |
|---|---|---|---|---|---|---|---|---|
| Origins | 4k±promoters | 20,250 | 26,237 | 4275 | 1000 | 649 | 6.6 | <0.001 |
| Origins | ORC_peaks | 20,250 | 12,712 | 347 | 1000 | 54 | 6.4 | <0.001 |
| Origins | R-loop_zones | 20,250 | 59,176 | 2607 | 1000 | 299 | 8.7 | <0.001 |
| Origins | G-quadruplex | 20,250 | 1,444,095 | 8818 | 1000 | 2529 | 3.5 | <0.001 |
| Origins | CTCF_sites | 20,250 | 22,097 | 443 | 1000 | 104 | 4.3 | <0.001 |
| Promoters | ORC_peaks | 26,237 | 12,712 | 4832 | 1000 | 454 | 10.6 | <0.001 |
| Promoters | R-loop_zones | 26,237 | 59,176 | 10,068 | 1000 | 1404 | 7.2 | <0.001 |
| Promoters | G-quadruplex | 26,237 | 1,444,095 | 21,809 | 1000 | 17,242 | 1.3 | <0.001 |
| Promoters | CTCF_sites | 20,250 | 22,097 | 3644 | 1000 | 813 | 4.5 | <0.001 |

overlap between origins and G-quadruplexes may be secondary to the overlap between origins and promoters. However, the overlap of shared origins with G-quadruplexes is 3.5-fold above random while that of promoters with G-quadruplexes is only 1.3-fold above random, which would be consistent with the idea that G-quadruplexes have a role in specifying origins and the overlap is not simply due to the proximity of origins with promoters.

Similarly, CTCF-binding sites have been proposed to contribute to origin specification (*Emerson et al., 2022*). Here again, the 2.2% of shared origins that overlap with constitutive CTCF-binding sites is less than the 6.4% of promoters that overlap with CTCF-binding sites (*Figure 2e*). Here, though, the permutation test in *Table 2* reveals that the enrichment of origins near the CTCF-binding sites (4.3-fold above random background) is comparable to the enrichment of promoters near the same sites (4.5-fold). This may, thus, suggest that the overlap of the CTCF-binding sites with origins is secondary to the overlap of origins with promoters.

Since ORC is an early factor for initiating DNA replication, shared human origins should be proximate to the reproducible ORC-binding sites. The vast majority of ORC-binding sites are not proximate to the shared origins and conversely only about 6.4% of the shared origins are proximate (within 1 kb) to the reproducible (shared) ORC-binding sites (*Figure 3f*). Even with the most relaxed criteria, nearly 85% of the most reproducible origins in human cells are not proximate to any ORC-binding site (union, *Figure 3f*). This low level of proximity contrasts with the ~100% of yeast origins that are within 1 kb of a yeast ORC-binding site. Even when we examined cell line-specific origins with the ORC ChIP-seq datasets from the same cell lines, the co-localization between origins and ORC-binding sites remained poor (*Figure 3—figure supplement 1f*). Another study also noted that only 13% of SNS-seq origins in K562 cells are near the ORC2-binding sites (*Miotto et al., 2016*), but the authors suggested that this could occur if many SNS-seq sites are not real origins but arose from DNA breaks. Vast contamination of SNS-seq origins by sites of DNA breaks is discounted in our analysis because of the reproducibility of the shared origins in multiple labs, multiple lineages and multiple different techniques.

Instead of an unbiased determination of origins mapped by multiple groups to create a small set of reproducible origins, we could also empirically select a few well-curated origin datasets. Toward that end we used the core origins identified by lambda exonuclease SNS-seq (*Akerman et al., 2020*) and the Ini-seq2 origins identified by labeling nuclei in vitro to identify the earliest replicated parts of the genome (*Guilbaud et al., 2022*). The shared origins were only a small subset of these origin datasets (*Figure 5—figure supplement 3a and c*). The percentage of these origins that were near ORC was still far less than what we observed in yeast: 13.7% for core origins and 30.4% for Ini-seq2 origins (*Figure 5—figure supplement 3b and d*). The higher overlap with Ini-seq2 origins could be because in vitro initiation of replication in isolated nuclei in Ini-seq preferentially identifies origins in early replicating, euchromatic parts of the genome, areas that have already been reported to be more enriched for the ORC2-binding sites (*Miotto et al., 2016*).

There could be other explanations for the poor overlap of shared origins with the ORC-binding sites. We tested whether increasing the stringency of reproduction in SNS-seq data will produce more reproducible origins that are better co-localized with the ORC-binding sites. However, when we reduced the shared origins based on their reproduction by 30, 40, or 50 (out of 66) SNS-seq datasets, we did not see a marked improvement of their co-localization with currently known ORC-binding sites (*Figure 5—figure supplement 4a and b*) and MCM-binding sites (*Figure 5—figure supplement 4c and d*).

Finally, the permutation test in *Table 2* demonstrates that the co-localization of origins with ORC-binding sites was 6.5-fold (p<0.001) greater than random expectation but this was less than the co-localization of promoters with ORC-binding sites, 10.6-fold greater than random (p<0.001). Thus, the co-localization data cannot conclusively say whether ORC binding near an origin is because ORC specifies origins, or whether this is secondary to origins being co-localized near promoters. Note that the co-localization of ORC1-binding sites and origins in HeLa cells with highly active TSS has been noted in the past (*Dellino et al., 2013*). In human lymphoblast Raji cells, ORC was enriched in promoters and MCM was depleted from gene bodies (*Kirstein et al., 2021*). ORC1 has also been reported to bind to RNA near promoters and stimulate replication from such sites (*Mas et al., 2023*). Thus, our observation that ORC is highly enriched near promoters is expected, the main advance being that the random permutation test suggests that co-localization of TSS with ORC is more than the co-localization of shared origins with ORC.

The question arises, are our results contradicting past studies? To the best of our knowledge, no one has taken such a comprehensive and quantitative approach to assess the proximity of origins with the ORC-binding sites genome-wide. We determined the percentage of origins that are within 1 kb of ORC-binding sites genome-wide and did a random permutation test to test whether the observed overlap is greater than random expectation. This showed that only a small percentage of the origins are within 1 kb of the currently known ORC-binding sites on a genome-wide scale, although the selected browser shots as shown in *Figure 6* might suggest otherwise. Furthermore, even though origins were enriched near the ORC-binding sites relative to random expectation, this enrichment is not more than that of TSS near the ORC-binding sites.

Taken together, these results suggest three possibilities: (a) the ORC ChIP-seq or origin determination datasets in humans are very noisy and that the ChIP-seq data in particular fail to identify functional ORC-binding sites; (b) the MCM2-7 loaded at the ORC-binding sites move very far to initiate origins, farther than 1 kb from the ORC-binding sites; and (c) there are as yet unexplored mechanisms by which many of the most reproducible origins are specified, mechanisms that might include ORC-independent modes of origin specification, similar to the DnaA-independent modes of origin specification in bacteria that use R-loops or DNA breaks to initiate replication (*Itoh and Tomizawa, 1980*; *Leela et al., 2021*; *Goswami and Gowrishankar, 2022*).

There is significant evidence for possibility (b). MCM2-7 are either loaded far from the ORC-binding sites or move significant distances after loading in the *Xenopus* egg extracts (*Harvey and Newport, 2003*). In yeasts, there is evidence that MCM2-7 are pushed far away from the bodies of transcriptionally active genes by the RNA polymerase II and initiate replication at sites distant from where they are loaded on chromatin by ORC (*Gros et al., 2015*). However, it is worth noting that despite this, nearly 100% of the origins in yeast are within 1 kb of the ORC-binding sites (*Figure 3f*). Similar observations have been made in *Drosophila*, where cyclin E/cdk2 kinase activity promotes the loading of a vast excess of MCM2-7 on chromatin relative to ORC, and the MCM2-7 complexes move away from their loading sites due to the activity of the transcriptional apparatus (*Powell et al., 2015*).

Since most models of replication initiation propose that a stably bound MCM2-7 complex is converted into an active CMG helicase at the time of origin firing, we hoped that even if ORC-binding sites are not necessarily close to the shared origins, MCM2-7-binding sites will be more proximate to the shared origins in human cells. However, only 4.5% of shared origins are near any MCM2-7-binding site in human cells and again this is in contrast to the 96.5% of origins in yeast being near any MCM2-7-binding sites (*Figure 5g*). Even if we focus on the limited data from the same cell line (K562 or HCT116), only 3.8% (union origins) or 5.5% shared origins overlap with union MCM3-7-binding sites in K562 cells and 19.5% of union origins overlap with any MCM2-binding sites in HCT116 cells (*Figure 3—figure supplement 1b, c, and e*). As with ORC, we also did the analysis with two selected origin datasets – the core origins (*Akerman et al., 2020*) and the Ini-seq2 origins (*Guilbaud et al., 2022*) – and found that still a small percentage of these origins (6.6 and 10.7%, respectively) were near any MCM2-7-binding sites. Finally, a recent paper reported the binding sites of phosphorylated isoforms of MCM2 (*Thakur et al., 2022*), and since phosphorylation of MCM2 is a prerequisite for MCM2-7 being activated as a helicase, we asked whether any of the phospho-MCM2-binding sites showed better co-localization with the shared origins (*Figure 5—figure supplement 2*). Phospho-S108 MCM2 showed the best result among the phospho-isoforms tested, with 22% of the shared origins being proximate to the phospho-S108 MCM2-binding sites. Thus, phosphorylation on S108 of MCM2 may indeed mark MCM2-7 complexes that become active helicases that fire origins, but even then the co-localization was seen in a disappointingly low percentage of shared origins.

MCM2-7 ChIP-seq data is biased by a lot of noise as human MCM2-7 slides around after initial loading and due to contamination from the actively replicating CMG helicase that moves all over the genome (and pauses at many sites) during the S phase, but this should increase the number of MCM2-7-binding sites and not decrease the percentage of origins that are near the MCM2-7 sites. Better co-localization of shared origins with phospho-S108 MCM2 suggests that the ChIP methods can be improved to identify a small but critical pool of MCM2-7 that are engaged with the DNA stably in a way that permits origin specification. Alternatively, mapping of the sites bound to the active CMG helicase (MCM2-7, CDC45, and GINS), before the active helicase moves too far from the initiation site, may better enrich sites that are destined to be origins.

A by-product of our analysis of ORC ChIP-seq data was the discovery that ORC1 and ORC2 are not strictly co-located as expected for the subunits of one ORC complex. Only 1363 (21%) of the 6501 ORC1 peaks overlapped with the 29,930 ORC2 peaks (*Figure 5—figure supplement 1a*), and the vast majority (68%) of ORC1 peaks are far away (≥2 kb) from the closest ORC2 peaks (*Figure 5—figure supplement 1b*). As a positive control, we checked other proteins that form well-established complexes like SMARCA4 and ARID1A in SWI/SNF complex, SMC1A and SMS63 in cohesion complex, EZH2 and SUZ12 in PRC2 complex, and found that they show a high proportion of overlap between their peaks as expected (*Figure 5—figure supplement 1a*). A possible explanation of the low overlapping rate between ORC1 and ORC2 peaks could be that the datasets are from different cell types (*Figure 5—figure supplement 1c*). However, even in ChIP-seq peaks from different cell types, we found continued high overlap between SMC1A and SMC3 peaks as would be expected for complexes that do not have high inter-cell-line variability in binding sites. In contrast, there was a low overlap between EZH2 and SUZ12 peaks taken from different cell lines, suggesting that for other complexes there is significant inter-cell-line variability in binding sites (*Figure 5—figure supplement 1d*). Thus, the lack of overlap between the ORC1 and ORC2 association sites on chromatin could be explained by inter-cell-line variation of binding sites for ORC (like the PRC2 complex) or could be explained by the ORC subunits binding to chromatin independent of one another. We prefer the latter explanation because of our experimental result that a substantial fraction of ORC2, 3, 4, 5, and 6 bind to chromatin in the absence of ORC1, and ORC2, 3, and 6 bind to chromatin normally in the absence of ORC5 (*Park and Asano, 2008*; *Shibata et al., 2016*; *Okano-Uchida et al., 2018*; *Shibata and Dutta, 2020*). We and others have also noted that human ORC2, 3, 4, and 5 form a stable subcomplex, with much looser association of the subcomplex with ORC1 and ORC6 (*Dhar et al., 2001*). Note that lineage-specific variation of where ORC may bind to the chromatin or the binding of ORC subcomplexes to chromatin may explain why so few of the ORC-binding sites overlap with the shared origins but does not explain why only 6.4% of the shared origins (identified reproducibly in different cell lines) is proximate to any ORC-binding sites.

Our analysis provides a characterization of origins in the human genome using multisource data in the public domain. The sequencing-based profiles from different techniques have different biases in detecting DNA replication origins and show different genomic features. As one adds criteria to identify the most reproducible experimentally determined origins that are close to the ORC-binding sites, the overlap with TSS, TF hotspots, constitutive CTCF-binding sites, and G-quadruplexes increases (*Table 1*), but in general the overlap with the latter three genomic features does not exceed that of the same features with TSS. Thus, although the overlaps are statistically significant, this correlation analysis cannot determine whether the same genomic features *independently* specify the origins of replication and promoters of transcription. This is a question that should be asked experimentally, and the 74 experimentally reproduced origin datasets with proximity to ORC and MCM2-7 ChIP-seq sites provide a starting list for such experimental queries. Finally, although our datasets for origins were large, as with all integrative data analyses, we are limited by the quality of the data, which can be improved as experimental techniques for both origin identification and ORC- or MCM2-7-binding site determination continue to improve.

## Methods
### Data processing
We collected all publicly available human sequencing datasets of replication origin profiling and ORC ChIP-seq from the GEO database (*Barrett et al., 2013*). Data details of the collected datasets can be found in *Supplementary file 1*. Raw sequence data in fastq format were downloaded and processed as follows: FastQC (v0.11.5; *Andrews, 2010*) was used for quality control and sequence data were then mapped to human genome (hg38) using bowtie2 (v2.2.9; *Langdon, 2015*). Sam files were converted into bam files using samtools (v1.12; *Li et al., 2009*) and only high-quality reads (q-score ≥ 30) were retained for subsequent analyses. Peak calling for ORC ChIP-seq data was performed with MACS2 (v2.1.4) callpeak function (--nomodel –extsize 150 -g hs -B –SPMR -q 0.05 –keep-dup 1) (*Zhang et al., 2008*). Samples with more than 1000 peaks were kept as high-quality samples. SNS-seq, Bubble-seq, and Rerep-seq peak calling was performed with SICER (*Zang et al., 2009*) with different parameters based on the different resolutions of technique: SNS-seq (W200 G600),

Bubble-seq (W5000 G15000), and Rerep-seq (W200 G600). The OK-seq-defined Izs were generated from previously published papers (*Petryk et al., 2016*; *Wu et al., 2018*), and the coordinate information was provided by Drs. Chunlong Chen and Olivier Hyrien (https://github.com/CL-CHEN-Lab/OK-Seq/tree/master/published_results). Repli-seq-defined origins were generated from the UW Encode group and are accessible in GSE34399. To obtain union peaks, peaks from each and all techniques were merged using Bedtools (v2.29.2; *Quinlan and Hall, 2010*) merge. Union origins that are longer than 300 bp were cut into separate origins of a maximum length of 300 bp. In the end, we obtained 7,459,709 union origins.

To compare the locations of origins between samples, we performed a PCA, where the matrix (to which PCA is applied) is of size 7.5M (number of union origins) × 113 (samples), with a '1' in the ($I$,$j$) position if the $i$th origin was detected in the $j$th dataset. We also performed pairwise Pearson correlation of the origins in all the datasets to determine the reproducibility of the results from a given technique and across techniques. The distance metric for heatmap is generated in R using the function cor (with parameters: use = "pairwise.complete.obs", method = "pearson").

## Identification of shared origins

The origins detected by each sample are very different, which makes the identification of common origins difficult. SNS-seq origins have the highest resolution, and so we start with them. We identified 5,069,432 union SNS-seq origins using the same approach. We focus on the union SNS-seq origins that appear above a threshold number of SNS-seq origin datasets (high-confidence SNS-seq origins) and then determine those that overlap with any IZ identified by each of the three different techniques to delineate a set of 'shared origins', which initiate replication all the time and can be detected by multiple approaches. Note that this approach does not lose the high resolution of the SNS-seq origins, but it merely finds those SNS-seq origins that are most reproduced and detected by each of the other three techniques. Rerep-seq is not included in defining shared origins because it is very different from the other techniques in many ways (*Figure 1b–h*, *Figure 1—figure supplement 1g*).

To identify the most commonly shared origins between SNS-seq samples, we use an occupancy score for each union SNS-seq origin to show how many samples identify that specific origin. A higher occupancy score means the origin is more commonly present in many samples (*Figure 2—figure supplement 1a and b*). High-confidence SNS-seq origins are those that occur in a sufficient number of SNS-seq samples at a false discovery rate (FDR) of >0.1. We used an exponential model for the background (green line) and plotted the distribution of occupancy scores (blue dotted curve) for origins from all SNS-seq samples (*Figure 2a*). The exponential model can be expressed as

$$y = A * e^{(K*(x+m))}$$

where $x$ is the occupancy score, and $y$ is the expected number of origins. Origins that are reproduced by multiple samples have higher occupancy scores than the background distribution. An empirical FDR of 0.1 was used to determine the cutoff of occupancy score so that the number of observed origins with occupancy score greater than the cutoff should be 10 times more than expected in the background model (*Fang et al., 2020*). The high-confidence SNS-seq origins were thus those with occupancy score ≥20 in all 66 SNS-seq samples. Detailed parameters can be found in *Supplementary file 2*. Origins mapping to ChrM are removed from SNS-seq shared origins to avoid the interference of mitochondria DNA.

Finally, the high-confidence SNS-seq origins defined by our model that overlap with any IZs from OK-seq, any IZs from Bubble-seq, and any origins defined by Repli-seq are called 'shared origins' (*Figure 2b*). A total of 20,250 shared origins were identified.

K562 shared origins were identified as K562 SNS-seq origins that overlapped with any of the union OK-seq and Repli-seq peaks. HeLa shared origins were identified as SNS-seq origins present in all three SNS-seq samples in HeLa and overlapped with any of the union OK-seq and Repli-seq peaks.

## Origins, ORC, and MCM ChIP-seq data in yeast

Yeast origins data were obtained from OriDB (*Nieduszynski et al., 2007*). A total of 829 replication origins mapped to the yeast genome sacCer1 were converted into the sacCer3 genome using UCSC LiftOver (*Hinrichs et al., 2006*). A total of 289 experimentally confirmed origins, considered as high-confidence origins, were used for co-localization analysis with ORC- and MCM-binding sites in yeast.

ORC and MCM ChIP-seq datasets were collected from the public domain (*Supplementary file 1*) and processed using the same procedure as used for human data with reference genome version sacCer3. The shared MCM-binding sites were defined as the MCM peaks that occur in all samples.

### Enrichment of histone marks at shared origins

Histone modification ChIP-seq peak files were downloaded from CistromeDB (*Zheng et al., 2019*). A total of 5711 peak files, each with over 5000 peaks, covering 29 distinct histone modifications in different human cell types were used to interrogate the association between histone marks and shared origins using union origins as control. The enrichment analysis was applied for each peak file by comparing the peak number in each histone modification peak file overlapping with the shared origins versus the overlapping with the union origins. The odds ratio obtained from two-tailed Fisher's exact test was used as the enrichment score for each peak file (*Figure 2j*). The odds ratio and p-values were calculated using the Python package scipy. (stats.fisher_exact([[shared_ori_overlap_with_hm, shared_ori_not_overlap_with_hm], [all_ori_overlap_with_hm - shared_ori_overlap_with_hm, all_ori_not_overlap_with_hm - shared_ori_not_overlap_with_hm]])).

### Co-localization analysis

We used co-localization analysis to define shared origins and show the overlapping of CTCF peaks, TF hotspots, TSS regions, G-quadruplex motifs, etc., with origins. The co-localization analysis was performed using Bedtools (*Quinlan and Hall, 2010*) intersect -u. At least 1 base pair of intersection is required to be defined as overlapping.

### Distance to the closest feature

According to the fact that ORC usually binds to origins but the loaded MCM2-7 can shift before firing an origin, we defined ORC that binds near the origins ±1 kb as ORC near origins. Bedtools closest -d -t "first" was used to find the closest peak/region and distance for a given region. Origins with a binding site of ORC no further than 1 kb were selected as origins near the ORC-binding site.

### G4 sites

G4 (G-quadruplex)-predicted motif sites were obtained from a published G4 motif predictor: G4Hunter (*Bedrat et al., 2016*). The 1,444,095 G4 motif coordinates in hg38 were provided by Laurent.

### Genomic annotation of promoter, exon, intron, and intergenic region coverage

The coordinates of TSS, exon, and intron are from UCSC hg38 version (*Navarro Gonzalez et al., 2021*). Promoter regions are defined as TSS ±1 kb. Intergenic regions are other genome regions excluding promoter, exon, and intron regions. Genomic annotation for peaks is defined by overlapping with those four types of regions by at least 1 bp.

### Replication timing score

We used publicly available Repli-seq data in K562 cell line for different cell phases from the ENCODE project (*Navarro Gonzalez et al., 2021*) to measure the replication timing of origins (*Supplementary file 1*). We used a previously defined weighted average score (*Hansen et al., 2010*) to combine the signal from the six-cell phases with the following formula: score = (0.917 * G1b) + (0.750 * S1) + (0.583 * S2) + (0.417 * S3) + (0.250 * S4) + (0 * G2). Higher values correspond to earlier replication.

### Permutation test

The R (version 4.1.3) package named regioneR (version 1.24.0) was used to statistically evaluate the associations between region sets with minor modifications. The following parameters were used for running regioneR. Number of iterations: 10,000. Evaluation function: numOverlaps. Randomization function: randomizeRegions. randomizeRegions: hg38. non.overlapping: TRUE. mc.set.seed: FALSE.

## Acknowledgements

This work was supported by R01 CA60499 to AD, R35 GM133712 to CZ, and K99 CA259526 to ZS. We thank all members of both the Zang and Dutta Labs for many helpful suggestions.

## Additional information

### Funding

| Funder | Grant reference number | Author |
|---|---|---|
| National Institutes of Health | R01 CA60499 | Anindya Dutta |
| National Institutes of Health | R35 GM133712 | Chongzhi Zang |
| National Institutes of Health | K99 CA259526 | Zhangli Su |

The funders had no role in study design, data collection and interpretation, or the decision to submit the work for publication.

### Author contributions

Mengxue Tian, Data curation, Software, Formal analysis, Investigation, Visualization, Methodology, Writing – original draft, Writing – review and editing; Zhenjia Wang, Data curation, Software, Formal analysis, Investigation, Methodology; Zhangli Su, Formal analysis, Investigation, Visualization, Methodology, Writing – review and editing; Etsuko Shibata, Resources, Investigation, Writing – review and editing; Yoshiyuki Shibata, Formal analysis, Methodology, Writing – review and editing; Anindya Dutta, Conceptualization, Resources, Formal analysis, Supervision, Funding acquisition, Validation, Investigation, Visualization, Writing – original draft, Project administration, Writing – review and editing; Chongzhi Zang, Conceptualization, Supervision, Funding acquisition, Investigation, Visualization, Methodology, Writing – original draft, Project administration, Writing – review and editing

### Author ORCIDs

Mengxue Tian (ID) https://orcid.org/0000-0003-3707-4225
Zhangli Su (ID) http://orcid.org/0000-0001-6495-3351
Anindya Dutta (ID) http://orcid.org/0000-0002-4319-0073
Chongzhi Zang (ID) http://orcid.org/0000-0003-4812-3627

Reviewer #1 (Public Review): https://doi.org/10.7554/eLife.89548.4.sa1
Reviewer #2 (Public Review): https://doi.org/10.7554/eLife.89548.4.sa2
Reviewer #3 (Public Review): https://doi.org/10.7554/eLife.89548.4.sa3
Author Response https://doi.org/10.7554/eLife.89548.4.sa4

## Additional files

### Supplementary files

• Supplementary file 1. Collected public data. The original reference for each dataset can be found on the GEO page for each dataset. (a) Metadata of collected public Origin data. (b) Sample number of each cell type. (c) Metadata of collected public ORC ChIP-seq data. (d) Metadata of collected public MCM ChIP-seq data. (e) Metadata of collected public ORC and MCM ChIP-seq data in yeast.

• Supplementary file 2. Parameters of model for SNS-seq. Parameters of exponential model for SNS-seq origins.

• Supplementary file 3. Union and shared ORC-binding sites. (a) Union ORC ChIP-seq peaks, with coordinates and occupancy scores. (b) Shared ORC-binding sites (defined as union ORC ChIP-seq peaks with an occupancy score ≥ 2).

• Supplementary file 4. Highest confidence origins. (a) Coordinates of union ORC ChIP-seq-binding sites with shared origins in 1 kb region. (b) Coordinates of shared origins with shared ORC-binding sites in 1 kb region. (c) Coordinates of shared origins with union ORC-binding sites in 1 kb region.

• Supplementary file 5. Seventy-four most confident origins. Coordinates of 74 origins that were reproducibly identified by multiple methods (shared origins, near shared ORC-binding sites, overlapping with MCM3-7-binding sites, and MCM2-binding sites).

• MDAR checklist

## Data availability

The current manuscript is a computational study, so no data have been generated for this manuscript. A summary of the collected public data can be found in *Supplementary file 1*. Modelling code and origin data are uploaded in: https://github.com/tmx1228/Replication-Origins (copy archived at *Tian, 2023*).

The following previously published datasets were used:

| Author(s) | Year | Dataset title | Dataset URL | Database and Identifier |
|---|---|---|---|---|
| Valsakumar V, Mesner LD, Pickin RR, Cieslik MP, Hamlin JL, Bekiranov S | 2013 | Bubble-seq analysis of the human genome reveals distinct chromatin-mediated mechanisms for regulating early- and late-firing origins | https://www.ncbi.nlm.nih.gov/geo/query/acc.cgi?acc=GSE38809 | NCBI Gene Expression Omnibus, GSE38809 |
| Murai J, Tang SW, Leo E, Baechler SA | 2018 | SLFN11 blocks stressed replication forks independently of ATR (NS-Seq) | https://www.ncbi.nlm.nih.gov/geo/query/acc.cgi?acc=GSE101515 | NCBI Gene Expression Omnibus, GSE101515 |
| Jang SM, Zhang Y, Utani K, Fu H | 2018 | The Replication-initiation determinant protein (RepID) modulates replication by recruiting CUL4 to chromatin | https://www.ncbi.xyz/geo/query/acc.cgi?acc=GSE114703 | NCBI Gene Expression Omnibus, GSE114703 |
| Akerman I, Kasaai B, Bazarova A, Sang PB | 2020 | A predictable conserved DNA base composition signature defines human core DNA replication origins | https://www.ncbi.nlm.nih.gov/geo/query/acc.cgi?acc=GSE128477 | NCBI Gene Expression Omnibus, GSE128477 |
| Pongor LS, Gross JM, Vera Alvarez R, Murai J | 2020 | BAMscale: quantification of DNA sequencing peaks and generation of scaled coverage tracks | https://www.ncbi.nlm.nih.gov/geo/query/acc.cgi?acc=GSE131417 | NCBI Gene Expression Omnibus, GSE131417 |
| Long H, Zhang L, Lv M, Wen Z | 2019 | H2A.Z Facilitates Licensing and Activation of Early Replication Origins | https://www.ncbi.nlm.nih.gov/geo/query/acc.cgi?acc=GSE134988 | NCBI Gene Expression Omnibus, GSE134988 |
| Fu H, Redon CE, Thakur BL, Utani K | 2021 | Dynamics of replication origin over-activation | https://www.ncbi.nlm.nih.gov/geo/query/acc.cgi?acc=GSE172417 | NCBI Gene Expression Omnibus, GSE172417 |
| Thakur BL, Baris AM, Fu H, Redon CE | 2022 | Convergence of SIRT1 and ATR signaling to modulate replication origin dormancy | https://www.ncbi.nlm.nih.gov/geo/query/acc.cgi?acc=GSE184353 | NCBI Gene Expression Omnibus, GSE184353 |
| Guilbaud G, Murat P, Wilkins H, Lerner LK, Sale JE, Krude T | 2022 | Determination of human DNA replication origin position and efficiency reveals principles of initiation zone organisation | https://www.ncbi.nlm.nih.gov/geo/query/acc.cgi?acc=GSE186675 | NCBI Gene Expression Omnibus, GSE186675 |
| Wu X, Kabalane H, Kahli M, Petryk N, Laperrousaz B, Jaszczyszyn Y | 2018 | Developmental and cancer-associated plasticity of DNA replication preferentially targets GC-poor, lowly expressed and late-replicating regions | https://www.ebi.ac.uk/ena/browser/view/PRJEB25180 | The European Nucleotide Archive, PRJEB25180 |

*Continued on next page*

*Continued*

| Author(s) | Year | Dataset title | Dataset URL | Database and Identifier |
|---|---|---|---|---|
| Petryk N, Kahli M, d'Aubenton-Carafa Y, Jaszczyszyn Y, Shen Y, Silvain M | 2016 | Replication landscape of the human genome | https://github.com/CL-CHEN-Lab/OK-Seq/tree/master/published_results | GitHub, CL-CHEN-Lab/OK-Seq/tree/master/published_results |
| Menzel J, Tatman P, Black JC | 2020 | Isolation and analysis of rereplicated DNA by Rerep-seq | https://www.ncbi.nlm.nih.gov/geo/query/acc.cgi?acc=GSE143572 | NCBI Gene Expression Omnibus, GSE143572 |
| Consortium ENCODE Project | 2012 | Replication Timing by Repli-seq from ENCODE/University of Washington | https://www.ncbi.nlm.nih.gov/geo/query/acc.cgi?acc=GSE34399 | NCBI Gene Expression Omnibus, GSE34399 |
| Dellino GI, Cittaro D, Piccioni R, Luzi L | 2012 | Genome-wide mapping of human DNA-replication origins: levels of transcription at ORC1 sites regulate origin selection and replication timing | https://www.ncbi.nlm.nih.gov/geo/query/acc.cgi?acc=GSE37583 | NCBI Gene Expression Omnibus, GSE37583 |
| Miotto B, Ji Z, Struhl K | 2016 | Selectivity of ORC binding sites and the relation to replication timing, fragile sites, and deletions in cancers | https://www.ncbi.nlm.nih.gov/geo/query/acc.cgi?acc=GSE70165 | NCBI Gene Expression Omnibus, GSE70165 |
| Utani K, Fu H, Jang SM, Marks AB | 2017 | Phosphorylated SIRT1 associates with replication origins to prevent excess replication initiation and preserve genomic stability | https://www.ncbi.nlm.nih.gov/geo/query/acc.cgi?acc=GSE94403 | NCBI Gene Expression Omnibus, GSE94403 |
| Ivanov MP, Ekker H, Peters J | 2017 | Genome-wide map of MCM3 and GFP-tagged ESCO2 mutants' binding in human fibroblast cell line vu1199-F SV40 | https://www.ncbi.nlm.nih.gov/geo/query/acc.cgi?acc=GSE80989 | NCBI Gene Expression Omnibus, GSE80989 |
| Lister R, Pelizzola M, Dowen RH, Hawkins RD | 2009 | UCSD Human Reference Epigenome Mapping Project | https://www.ncbi.nlm.nih.gov/geo/query/acc.cgi?acc=GSE16256 | NCBI Gene Expression Omnibus, GSE16256 |
| Gros J, Kumar C, Lynch G, Yadav T | 2016 | Post-licensing specification of eukaryotic replication origins by facilitated Mcm2-7 sliding along DNA | https://www.ncbi.nlm.nih.gov/geo/query/acc.cgi?acc=GSE69065 | NCBI Gene Expression Omnibus, GSE69065 |
| Eaton ML, Galani K, Kang S, Bell SP | 2010 | ORC precisely positions nucleosomes at origins of replication | https://www.ncbi.nlm.nih.gov/geo/query/acc.cgi?acc=GSE16926 | NCBI Gene Expression Omnibus, GSE16926 |
| CSK Lee, Cheung MF, Li J, Zhao Y | 2020 | Humanizing the Yeast Origin Recognition Complex | https://www.ncbi.nlm.nih.gov/geo/query/acc.cgi?acc=GSE149163 | NCBI Gene Expression Omnibus, GSE149163 |

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
