## [Editor Report · eLife assessment]

The article addresses the mechanism of initiation of DNA replication in human cells by analyzing published data on the location of origins of DNA replication and the location of binding sites in the genome for ORC and MCM2-7 complexes. There are some **useful** analyses of existing data but there are concerns regarding the conclusion that there might be alternative mechanisms for determining the location of origins of DNA replication in human cells compared to the well-known mechanism known from many eukaryotic systems, including yeast, *Xenopus*, *C. elegans,* and *Drosophila*. The lack of overlap between binding sites for ORC1 and ORC2, which are known to form a complex in human cells, is a particular concern and points to the evidence for the accurate localization of their binding sites in the genome being **incomplete**.

---

## [Referee Report · Reviewer #1 (Public Review)]

In the best genetically and biochemically understood model of eukaryotic DNA replication, the budding yeast, *Saccharomyces cerevisiae*, the genomic locations at which DNA replication initiates are determined by a specific sequence motif. These motifs, or ARS elements, are bound by the origin recognition complex (ORC). ORC is required for loading of the initially inactive MCM helicase during origin licensing in G1. In human cells, ORC does not have a specific sequence binding domain and origin specification is not specified by a defined motif. There have thus been great efforts over many years to try to understand the determinants of DNA replication initiation in human cells using a variety of approaches, which have gradually become more refined over time.

In this manuscript Tian et al. combine data from multiple previous studies using a range of techniques for identifying sites of replication initiation to identify conserved features of replication origins and to examine the relationship between origins and sites of ORC binding in the human genome. The authors identify (a) conserved features of replication origins e.g. association with GC-rich sequences, open chromatin, promoters and CTCF binding sites. These associations have already been described in multiple earlier studies. They also examine the relationship of their determined origins and ORC binding sites and conclude that there is no relationship between sites of ORC binding and DNA replication initiation. While the conclusions concerning genomic features of origins are not novel, if true, a clear lack of colocalization of ORC and origins would be a striking finding. However, the majority of the datasets used do not report replication origins, but rather broad zones in which replication origins fire. Rather than refining the localisation of origins, the approach of combining diverse methods that monitor different objects related to DNA replication leads to a base dataset that is highly flawed and cannot support the conclusions that are drawn, as explained in more detail below.

Methods to determine sites at which DNA replication is initiated can be divided into two groups based on the genomic resolution at which they operate. Techniques such as bubble-seq, ok-seq can localise zones of replication initiation in the range ~50kb. Such zones may contain many replication origins. Conversely, techniques such as SNS-seq and ini-seq can localise replication origins down to less than 1kb. Indeed, the application of these different approaches has led to a degree of controversy in the field about whether human replication does indeed initiate at discrete sites (origins), or whether it initiates randomly in large zones with no recurrent sites being used. However, more recent work has shown that elements of both models are correct i.e. there are recurrent and efficient sites of replication initiation in the human genome, but these tend to be clustered and correspond to the demonstrated initiation zones (Guilbaud et al., 2022).

These different scales and methodologies are important when considering the approach of Tian et al. The premise that combining all available data from five techniques will increase accuracy and confidence in identifying the most important origins is flawed for two principal reasons. First, as noted above, of the different techniques combined in this manuscript, only SNS-seq can actually identify origins rather than initiation zones. It is the former that matters when comparing sites of ORC binding with replication origin sites, if a conclusion is to be drawn that the two do not co-localise.

Second, the authors give equal weight to all datasets. Certainly, in the case of SNS-seq, this is not appropriate. The technique has evolved over the years and some earlier versions have significantly different technical designs that may impact the reliability and/or resolution of the results e.g. in Foulk et al. (Foulk et al., 2015), lambda exonuclease was added to single stranded DNA from a total genomic preparation rather than purified nascent strands, which may lead to significantly different digestion patterns (ie underdigestion). Curiously, the authors do not make the best use of the largest SNS-seq dataset (Akerman et al., 2020) by ignoring these authors separation of core and stochastic origins. By blending all data together any separation of signal and noise is lost. Further, I am surprised that the authors have chosen not to use data and analysis from a recent study that provides subsets of the most highly used and efficient origins in the human genome, at high resolution (Guilbaud et al., 2022).

References

Akerman I, Kasaai B, Bazarova A, Sang PB, Peiffer I, Artufel M, Derelle R, Smith G, Rodriguez-Martinez M, Romano M, Kinet S, Tino P, Theillet C, Taylor N, Ballester B, Méchali M (2020) A predictable conserved DNA base composition signature defines human core DNA replication origins. Nat Commun, 11: 4826

Foulk MS, Urban JM, Casella C, Gerbi SA (2015) Characterizing and controlling intrinsic biases of lambda exonuclease in nascent strand sequencing reveals phasing between nucleosomes and G-quadruplex motifs around a subset of human replication origins. Genome Res, 25: 725-735

Guilbaud G, Murat P, Wilkes HS, Lerner LK, Sale JE, Krude T (2022) Determination of human DNA replication origin position and efficiency reveals principles of initiation zone organisation. Nucleic Acids Res, 50: 7436-7450

Update in response to authors' comments on the original review:

While the authors have clarified their approach to some aspects of their analysis, I believe they and I are just going to have to disagree about the methodology and conclusions of this work. I do not find the authors responses sufficiently compelling to change my mind about the significance of the study or veracity of the conclusions. In my opinion, the method for identification of strong origins is not robust and of insufficient resolution. In addition, the resolution and the overlap of the MCM Chip-seq datasets is poor. While the conclusion of the paper would indeed be striking and surprising if true, I am not at all persuaded that it is based on the presented data.

---

## [Referee Report · Reviewer #2 (Public Review)]

Tian et al. performed a meta-analysis of 113 genome-wide origin profile datasets in humans to assess the reproducibility of experimental techniques and shared genomics features of origins. Techniques to map DNA replication sites have quickly evolved over the last decade, yet little is known about how these methods fare against each other (pros and cons), nor how consistent their maps are. The authors show that high-confidence origins recapitulate several known features of origins (e.g., correspondence with open chromatin, overlap with transcriptional promoters, CTCF binding sites). However, surprisingly, they find little overlap between ORC/MCM binding sites and origin locations.

Overall, this meta-analysis provides the field with a good assessment of the current state of experimental techniques and their reproducibility, but I am worried about: (a) whether we've learned any new biology from this analysis; (b) how binding sites and origin locations can be so mismatched, in light of numerous studies that suggest otherwise; and (c) some methodological details described below.

-- I understand better the inclusion/exclusion logic for the samples. But I'm still not sure about the fragments. As the authors wrote, there is both noise and stochasticity; the former is not important but the latter is essential to include. How can these two be differentiated, and what may be the expected overlap as a function of different stochasticity rates?

-- Many of the major genomic features analyzed have already been found to be associated with origin sites. For example, the correspondence with TSS has been reported before:

https://www.ncbi.nlm.nih.gov/pmc/articles/PMC6320713/

https://www.ncbi.nlm.nih.gov/pmc/articles/PMC6547456/

-- Line 250: The most surprising finding is that there is little overlap between ORC/MCM binding sites and origin locations. The authors speculate that the overlap between ORC1 and ORC2 could be low because they come from different cell types. Equally concerning is the lack of overlap with MCM. If true, these are potentially major discoveries that butts heads with numerous other studies that have suggested otherwise.

The key missing dataset is ORC1 and ORC2 CHiP-seq from the same cell type. This shouldn't be too expensive to perform, and I hope someone performs this test soon. Without this, I remain on the fence about how much existing datasets are "junk" vs how much the prevailing hypothesis about replication needs to be revisited. Nonetheless, the authors do perform a nice analysis showing that existing techniques should be carefully used and interpreted.

---

## [Referee Report · Reviewer #3 (Public Review)]

Summary: The authors present a thought-provoking and comprehensive re-analysis of previously published human cell genomics data that seeks to understand the relationship between the sites where the Origin Recognition Complex (ORC) binds chromatin, where the replicative helicase (Mcm2-7) is loaded, and where DNA replication actually begins (origins). The view that these should coincide is influenced by studies in yeast where ORC binds site-specifically to dedicated nucleosome-free origins where Mcm2-7 can be loaded and remains stably positioned for subsequent replication initiation. However, this is most certainly not the case in metazoans where it has already been reported that chromatin bindings sites of ORC and Mcm2-7 do not necessarily overlap, nor do they always overlap with origins. This is likely due to Mcm2-7 possessing linear mobility on DNA (i.e., it can slide) such that other chromatin-contextualized processes can displace it from the site in which it was originally loaded. Additionally, Mcm2-7 is loaded in excess and thus only a fraction of Mcm2-7 would be predicted to coincide with replication start sites. This study reaches a very similar conclusion of these previous studies: they find a high degree of discordance between ORC, Mcm2-7, and origin positions in human cells.

Strengths: The strength of this work is its comprehensive and unbiased analysis of all relevant genomics datasets. To my knowledge, this is the first attempt to integrate these observations. It also is an important cautionary tale to not confuse replication factor binding sites with the genomic loci where replication actually begins, although this point is already widely appreciated in the field.

Weaknesses: The major weakness of this paper is the lack of novel biological insight and that the comprehensive approach taken failed to provide any additional mechanistic insight regarding how and why ORC, Mcm2-7, and origin sites are selected or why they may not coincide.

---

## [Author Response]

The following is the authors’ response to the previous reviews.

**eLife assessment**
The paper contains some useful analysis of existing data but there are concerns regarding the conclusion that there might be alternative mechanisms for determining the location of origins of DNA replication in human cells compared to the well known mechanism known from many eukaryotic systems, including yeast, *Xenopus*, *C. elegans* and *Drosophila*. The lack of overlap between binding sites for ORC1 and ORC2, which are known to form a complex in human cells, is a particular concern and points to the evidence for the accurate localization of their binding sites in the genome being incomplete.
**Public Reviews:**

**Reviewer #1 (Public Review):**
In the best genetically and biochemically understood model of eukaryotic DNA replication, the budding yeast, *Saccharomyces cerevisiae*, the genomic locations at which DNA replication initiates are determined by a specific sequence motif. These motifs, or ARS elements, are bound by the origin recognition complex (ORC). ORC is required for loading of the initially inactive MCM helicase during origin licensing in G1. In human cells, ORC does not have a specific sequence binding domain and origin specification is not specified by a defined motif. There have thus been great efforts over many years to try to understand the determinants of DNA replication initiation in human cells using a variety of approaches, which have gradually become more refined over time.In this manuscript Tian et al. combine data from multiple previous studies using a range of techniques for identifying sites of replication initiation to identify conserved features of replication origins and to examine the relationship between origins and sites of ORC binding in the human genome. The authors identify (a) conserved features of replication origins e.g. association with GC-rich sequences, open chromatin, promoters and CTCF binding sites. These associations have already been described in multiple earlier studies. They also examine the relationship of their determined origins and ORC binding sites and conclude that there is no relationship between sites of ORC binding and DNA replication initiation. While the conclusions concerning genomic features of origins are not novel, if true, a clear lack of colocalization of ORC and origins would be a striking finding. However, the majority of the datasets used do not report replication origins, but rather broad zones in which replication origins fire. Rather than refining the localisation of origins, the approach of combining diverse methods that monitor different objects related to DNA replication leads to a base dataset that is highly flawed and cannot support the conclusions that are drawn, as explained in more detail below.

Response: We are using the narrowly defined SNS-seq peaks as the gold standard origins and making sure to focus in on those that fall within the initiation zones defined by other methods. The objective is to make a list of the most reproducible origins. Unlike what the reviewer states, this actually refines the dataset to focus on the SNS origins that have also been reproduced by the other methods in multiple cell lines. We have changed the last box of Fig. 1A to make this clearer: Shared origins = reproducible SNS-seq origins that are contained in initiation zones defined by Repli-seq, OK-seq and Bubble-seq. This and the Fig. 2B (as it is) will make our strategy clearer.

Methods to determine sites at which DNA replication is initiated can be divided into two groups based on the genomic resolution at which they operate. Techniques such as bubble-seq, ok-seq can localise zones of replication initiation in the range ~50kb. Such zones may contain many replication origins. Conversely, techniques such as SNS-seq and ini-seq can localise replication origins down to less than 1kb. Indeed, the application of these different approaches has led to a degree of controversy in the field about whether human replication does indeed initiate at discrete sites (origins), or whether it initiates randomly in large zones with no recurrent sites being used. However, more recent work has shown that elements of both models are correct i.e. there are recurrent and efficient sites of replication initiation in the human genome, but these tend to be clustered and correspond to the demonstrated initiation zones (Guilbaud et al., 2022).These different scales and methodologies are important when considering the approach of Tian et al. The premise that combining all available data from five techniques will increase accuracy and confidence in identifying the most important origins is flawed for two principal reasons. First, as noted above, of the different techniques combined in this manuscript, only SNS-seq can actually identify origins rather than initiation zones. It is the former that matters when comparing sites of ORC binding with replication origin sites, if a conclusion is to be drawn that the two do not co-localise.

Response: We agree. So the reviewer should agree that our method of finding SNS-seq peaks that fall within initiation zones actually refines the origins to find the most reproducible origins. We are not losing the spatial precision of the SNS-seq peaks.

Second, the authors give equal weight to all datasets. Certainly, in the case of SNS-seq, this is not appropriate. The technique has evolved over the years and some earlier versions have significantly different technical designs that may impact the reliability and/or resolution of the results e.g. in Foulk et al. (Foulk et al., 2015), lambda exonuclease was added to single stranded DNA from a total genomic preparation rather than purified nascent strands, which may lead to significantly different digestion patterns (ie underdigestion). Curiously, the authors do not make the best use of the largest SNS-seq dataset (Akerman et al., 2020) by ignoring these authors separation of core and stochastic origins. By blending all data together any separation of signal and noise is lost. Further, I am surprised that the authors have chosen not to use data and analysis from a recent study that provides subsets of the most highly used and efficient origins in the human genome, at high resolution (Guilbaud et al., 2022).

Response: (1) We are using the data from Akerman et al., 2020: Dataset GSE128477 in Supplemental Table 1. We have now separately examined the core origins defined by the authors to check its overlap with ORC binding (Supplementary Fig. S8b)

1. To take into account the refinement of the SNS-seq methods through the years, we actually included in our study only those SNS-seq studies after 2018, well after the lambda exonuclease method was introduced. Indeed, all 66 of SNS-seq datasets we used were obtained after the lambda exonuclease digestion step. To reiterate, we recognize that there may be many false positives in the individual origin mapping datasets. Our focus is on the True positives, the SNS-seq peaks that have some support from multiple SNS-seq studies AND fall within the initiation zones defined by the independent means of origin mapping (described in Fig. 1A and 2B). These True positives are most likely to be real and reproducible origins and should be expected to be near ORC binding sites.

We have changed the last box of Fig. 1A to make this clearer: Shared origins = reproducible SNS-seq origins that are contained in initiation zones defined by Repli-seq, OK-seq or Bubble-seq.

Ini-seq by Torsten Krude and co-workers (Guillbaud, 2022) does NOT use Lambda exonuclease digestion. So using Ini-seq defined origins is at odds with the suggestion above that we focus only on SNS-seq datasets that use Lambda exonuclease. However, Ini-seq identifies a much smaller subset of SNS-seq origins, so, as requested, we have also done the analysis with just that smaller set of origins, and it does show a better proximity to ORC binding sites, though even then the ORC proximate origins account for only 30% of the Ini-seq2 origins (Supplementary Fig. S8d). Note Ini-seq2 identifies DNA replication initiation sites seen in vitro on isolated nuclei.

Update in response to authors' comments on the original review:While the authors have clarified their approach to some aspects of their analysis, I believe they and I are just going to have to disagree about the methodology and conclusions of this work. I do not find the authors responses sufficiently compelling to change my mind about the significance of the study or veracity of the conclusions. In my opinion, the method for identification of strong origins is not robust and of insufficient resolution. In addition, the resolution and the overlap of the MCM Chip-seq datasets is poor. While the conclusion of the paper would indeed be striking and surprising if true, I am not at all persuaded that it is based on the presented data.
**Reviewer #2 (Public Review):**
Tian et al. performed a meta-analysis of 113 genome-wide origin profile datasets in humans to assess the reproducibility of experimental techniques and shared genomics features of origins. Techniques to map DNA replication sites have quickly evolved over the last decade, yet little is known about how these methods fare against each other (pros and cons), nor how consistent their maps are. The authors show that high-confidence origins recapitulate several known features of origins (e.g., correspondence with open chromatin, overlap with transcriptional promoters, CTCF binding sites). However, surprisingly, they find little overlap between ORC/MCM binding sites and origin locations.Overall, this meta-analysis provides the field with a good assessment of the current state of experimental techniques and their reproducibility, but I am worried about: (a) whether we've learned any new biology from this analysis; (b) how binding sites and origin locations can be so mismatched, in light of numerous studies that suggest otherwise; and (c) some methodological details described below.I understand better the inclusion/exclusion logic for the samples. But I'm still not sure about the fragments. As the authors wrote, there is both noise and stochasticity; the former is not important but the latter is essential to include. How can these two be differentiated, and what may be the expected overlap as a function of different stochasticity rates?

It is difficult to separate the effect of noise from the effect of stochastic firing of origins. We therefore took the simplest approach: focus only on the most reproducible origins (shared origins) and ignore the non-reproducible origins. At least the most reproducible origins can be used to test the hypotheses regarding origin firing.

Many of the major genomic features analyzed have already been found to be associated with origin sites. For example, the correspondence with TSS has been reported before:
https://www.ncbi.nlm.nih.gov/pmc/articles/PMC6320713/

https://www.ncbi.nlm.nih.gov/pmc/articles/PMC6547456/
Line 250: The most surprising finding is that there is little overlap between ORC/MCM binding sites and origin locations. The authors speculate that the overlap between ORC1 and ORC2 could be low because they come from different cell types. Equally concerning is the lack of overlap with MCM. If true, these are potentially major discoveries that butts heads with numerous other studies that have suggested otherwise.

The key missing dataset is ORC1 and ORC2 CHiP-seq from the same cell type. This shouldn't be too expensive to perform, and I hope someone performs this test soon. Without this, I remain on the fence about how much existing datasets are "junk" vs how much the prevailing hypothesis about replication needs to be revisited. Nonetheless, the authors do perform a nice analysis showing that existing techniques should be carefully used and interpreted.

We agree that a thorough set of ChIP-seq data (with multiple antibodies or with equivalent techniques that do not use antibodies) for all six subunits of ORC in mammalian cells will be very useful for the field. Note, though, that just by simple cell lysis, it is very easy to divide human ORC into at least three different parts: ORC1, ORC2-5, and ORC6. The subunits do not form as robust a complex as seen in the yeasts and in flies.

**Reviewer #3 (Public Review):**
Summary: The authors present a thought-provoking and comprehensive re-analysis of previously published human cell genomics data that seeks to understand the relationship between the sites where the Origin Recognition Complex (ORC) binds chromatin, where the replicative helicase (Mcm2-7) is loaded, and where DNA replication actually beings (origins). The view that these should coincide is influenced by studies in yeast where ORC binds site-specifically to dedicated nucleosome-free origins where Mcm2-7 can be loaded and remains stably positioned for subsequent replication initiation. However, this is most certainly not the case in metazoans where it has already been reported that chromatin bindings sites of ORC and Mcm2-7 do not necessarily overlap, nor do they always overlap with origins. This is likely due to Mcm2-7 possessing linear mobility on DNA (i.e., it can slide) such that other chromatin-contextualized processes can displace it from the site in which it was originally loaded. Additionally, Mcm2-7 is loaded in excess and thus only a fraction of Mcm2-7 would be predicted to coincide with replication start sites. This study reaches a very similar conclusion of these previous studies: they find a high degree of discordance between ORC, Mcm2-7, and origin positions in human cells.Strengths: The strength of this work is its comprehensive and unbiased analysis of all relevant genomics datasets. To my knowledge, this is the first attempt to integrate these observations. It also is an important cautionary tale to not confuse replication factor binding sites with the genomic loci where replication actually begins, although this point is already widely appreciated in the field.Response: Thank you for recognizing the comprehensive and unbiased nature of our analysis. Our findings will prevent the unwise adoption of ORC or MCM binding sites as surrogate markers of origins and will stimulate the field to try and improve methods of identifying ORC or MCM binding until the binding sites are found to be proximal to the most reproducible origins. The last possibility is that there are ORC- or MCM-independent modes of defining origins, but we have no evidence of that.Weaknesses: The major weakness of this paper is the lack of novel biological insight and that the comprehensive approach taken failed to provide any additional mechanistic insight regarding how and why ORC, Mcm2-7, and origin sites are selected or why they may not coincide.

Response: we agree that we cannot provide a novel biological insight from this kind of meta-analysis. The importance of this study is in highlighting that there is either significant problems with the data collected till now (preventing the co-localization of ORC or MCM binding sites with the most reproducible origins) or ORC and MCM binding sites are often far away from where the most reproducible origins fire, which should make us consider ways in which origins could be activated kilobases away from ORC and MCM binding sites.

**Recommendations for the authors:**

**Reviewer #2 (Recommendations For The Authors):**
All suggestions and recommendations were described in a previous review.
**Reviewer #3 (Recommendations For The Authors):**
The most significant omission is a contextualization of the results in the discussion and an explanation of why these results matter for the biology of replication, disease, and/or our confidence in the genomic techniques reported on in this study. As written, the discussion simply restates the results without any interpretation towards novel insight. I suggest that the authors revise their discussion to fill this important gap.A second important, unresolved point is whether replication origins identified by the various methods differ due to technical reasons or because different cell types were analyzed. Given the correlation between TSS and origins (reported in this study but many others too), it is somewhat expected that origins will differ between cell types as each will have a distinct transcriptional program. This critique is partly addressed in Figure S1C. However, given the conclusion that the techniques are only rarely in agreement (only 0.27% origins reproducibly detected by the four techniques), a more in-depth analysis of cell type specific data is warranted. Specifically, I would suggest that cell type-specific data be reported wherever origins have been defined by at least two methods in the same cell type, specifically reporting the percent of shared origins amongst the datasets. This type of analysis may also inform on whether one or more techniques produces the highest (or lowest) quality list of true origins.

We have done what has been suggested: used K562 cell type-specific data because here the origins have been defined by at least two methods in the same cell type and reported the percent of shared origins amongst the datasets (Supp. Fig. S4).

Other MINOR comments include:Line 215: the authors show that shared origins overlap with TF binding hotspots more often than union origins, which they claim suggests "that they are more likely to interact with transcription factors." As written, it sounds like the authors are proposing that ORC may have some direct physical interaction with transcription factors. Is this intended? If so, what support is there for this claim?

The reviewer is correct. We have rephrased because we have no experimental support for this claim.

In the text, Figure 3G is discussed before Figure 3F. I suggest switching the order of these panels in Figure 3.

Done.

It's not clear what Figure 5H to Figure 6 accomplishes. What specifically is added to the story by including these data? Is there something unique about the high confidence origins? If there is nothing noteworthy, I would suggest removing these data.

We want to keep them to highlight the small number of origins that meet the hypothesis that ORC and MCM must bind at or near reproducible origins. These would be the origins that the field can focus in on for testing the hypothesis rigorously. They also show the danger of evaluating proximity between ORC or MCM binding sites with origins based on a few browser shots. If we only showed this figure, we could conclude that ORC and MCM binding sites are very close to reproducible origins.

Line 394: "Since ORC is an early factor for initiating DNA replication, we expected that shared human origins will be proximate to the reproducible ORC binding sites." This is only expected if one disbelieves the prior literature that shows that ORC and origins are not, in many cases, proximal. This statement should be revised, or the previous literature should be cited, and an explanation provided about why this prior work may have missed the mark.

We do not know of any genome-wide study in mammalian cell lines where ORC binding sites and MCM binding have been compared to highly reproducible origins, or that show that these binding sites and highly reproducible origins are mostly not proximal to each other. Most studies cherry pick a few origins and show by ChIP-PCR that ORC and/or MCM bind near those sites. Alternatively, studies sometimes show a selected browser shot, without a quantitative measure of the overlap genome wide and without doing a permutation test to determine if the observed overlap or proximity is higher than what would be expected at random with similar numbers of sites of similar lengths. In the revised manuscript we have discussed Dellino, 2013; Kirstein, 2021; Wang, 2017; Mas, 2023. None of them have addressed what we are addressing, is the small subset of the most reproducible origins proximal to ORC or MCM binding sites?

Line 402-404: given the lack of agreement between ORC binding sites and origins the authors suggest as an explanation that "MCM2-7 loaded at the ORC binding sites move much further away to initiate origins far from the ORC binding sites, or that there are as yet unexplored mechanisms of origin specification in human cancer cells". The first part of this statement has been shown to be true (Mcm2-7 movement) and should be cited. But what do the authors mean by the second suggestion of "unexplored mechanisms"? Please expand.

We have addressed this point in the revised manuscript.

The authors should better reference and discuss the previous literature that relates to their work, some of these include Gros et al., 2015 Mol Cell, Powell et al., 2015 EMBO J, Miotto et al., 2016 PNAS, but likely there are many others.

We have addressed this point in the revised manuscript.

Note for authors:Line 107: The introduction discusses the mechanism for yeast ORC recognizes specific origins and discusses the Orc4 contribution, but it is known that Orc2 also binds DNA on a base-specific manner (see PMID 33056978). Thus Lee et al. did not "humanize ORC" as stated.

Done

Lines 117-119: Two of the cited papers are on endo-reduplication and not on initiation in a normal cell cycle and this should be pointed out. Second, there is contradictory evidence that ORC is essential in human cells and this should be cited (PMID 33522487)

Done